# Barriers of urban hydro-meteorological simulation: a review

Xuan Chen[1], Job Augustijn van der Werf[1], Arjan Droste[1], Miriam Coenders-Gerrits[1], and Remko Uijlenhoet[1]

[1]Department of Water Management, Faculty of Civil Engineering and Geosciences, Delft University of Technology, P.O Box 5048, 2600 GA Delft, the Netherlands

**Correspondence:** Xuan Chen (x.chen-12@tudelft.nl)

**Abstract.** Urban areas, characterized by dense populations and many socio-economic activities, increasingly suffer from floods, droughts, and heat stress due to land use and climate change. Traditionally, the urban thermal environment and water resources management have been studied separately, using urban land surface models (ULSMs) and urban hydrological models (UHMs). However, as our understanding deepens and the urgency to address future climate disasters grows, it becomes clear that hydroclimatological extremes—such as floods, droughts, severe urban thermal environments, and more frequent heat waves—are actually not always isolated events but can be compound events. This underscores the close interaction between the water cycle and the energy balance. Consequently, the existing separation between ULSMs and UHMs creates significant obstacles in better understanding urban hydrological and meteorological processes, which is crucial for addressing the high risks posed by climate change. Defining the future direction of process-based models for hydro-meteorological predictions and assessments is essential for better managing extreme events and evaluating response measures in densely populated urban areas. Our review focuses on three critical aspects of urban hydro-meteorological simulation: similarities, differences, and gaps among different models; existing gaps in physical process implementations; and efforts, challenges, and potential for model coupling and integration. We find that ULSMs inadequately represent water surfaces and hydraulic systems, while UHMs lack explicit surface energy balance solutions and detailed building representations. Coupled models show potential for simulating urban hydro-meteorological environments, but face challenges at regional and neighbourhood scales. Our review highlights the need for interdisciplinary communication between the urban climatology and the urban water management communities to enhance urban hydro-meteorological simulation models.

## Acronyms

**AH** Anthropogenic Heat

**BsoL** Bare Land

**BudF** Building Facet

**CFD** Computational Fluid Dynamics

**ImpL** Impervious Land

**LW** Longwave Radiation

**ML** Multilayer

**RBS** Runoff Branch Btructure

**RunoffG2G** Grid-to-grid Runof

**SL** Single Layer

**SW** Shortwave Radiation

  **UCMs** Urban Canopy Models

  **UHMs** Urban Hydrological Models

  **ULSMs** Urban Land surface Models

  **VegL** Urban Vegetated Land

**WatS** Water Surface

  **WRF** Weather Research and Forecast Models

## List of Symbols

$E$ Evaporation

$ET$ Evapotranspiration

$GH$ Ground Heat Flux

$H$ Sensible Heat Flux

$I$ Infiltration

$LE$ Latent Heat Flux

$LF_b$ Lateral Flow Between Grid Cells

$LF_{in}$ Lateral Flow Within Grid Cells

$Q_{atm}$ First Level Atmosphere Humidity

$Q_{can}$ Urban Canyon Humidity

$RH$ Relative Humidity

$R_{net}$ Net Radiation

$T_a$ Air Temperature

$T_{atm}$ First Level Atmosphere Temperature

$T_{can}$ Urban Canyon Air Temperature

$T_{gr}$ Green Roof Surface Temperature

$T_r$ Roof Surface Temperature

$T_s$ Surface Temperature

$T_t$ Tree Surface Temperature

$T_w$ Water Surface Temperature

VF Vertical Flow

$W_{can}$ Urban Canyon Wind

# 1   Introduction

Cities are characterised by dense populations and intense socio-economic activity, leading to intensive landscape modifications. Meteorology is notably intricate and heterogeneous in urban areas, and the hydrological processes differ significantly from those found in the natural environment. Although small-scale heterogeneity is common in many natural systems, urban areas display distinct patterns and scales of heterogeneity shaped by human activities, which in turn have a direct impact on people.

As a result, numerical simulations are often used in research to understand the impact of meteorological and hydrological processes on urban areas and enhance urban resilience in the face of future climate change. Process-based simulations can be used to predict urban climate dynamics and hydrological conditions, as well as assess the effects of urban expansion and the effectiveness of mitigation and response measures. Over the past decades, significant developments have been made in numerical meteorological and hydrological models tailored explicitly for urban environments (Fletcher et al., 2013; Lipson

et al., 2024). A systematic literature review will be provided later in this paper to cover the developments and advancements of these models.

To adequately simulate urban climate dynamics, land surfaces parameterisation schemes for urban land surface have been the main development in recent years, which may also include the urban land-atmosphere exchange. *Urban land surface models* (ULSMs) have been intensely applied to simulate urban thermal conditions in the past decades. The cornerstone of ULSMs

is resolving the urban surface energy balance, meaning that in literature, these have also been referred to as *urban surface energy balance models*. The main modelling philosophy is the same as that of the natural land surfaces models: the surface's effects on heat, moisture, and momentum fluxes need to be accounted for in the land-surface schemes used by numerical models. Existing models can exhibit varying levels of sophistication, depending on their underlying assumptions regarding surface features and exchange processes. To compare the capacity of the existing ULSMs, Grimmond et al. (2010) conducted the International Urban Energy Balance Models Comparison Project (PILPS-Urban). In this project, they summarised and compared thirty-three ULSMs. One of the conclusions drawn was that models have a good overall capability to model net all-wave radiation and a limited capability to model latent heat flux. Over a decade later, Urban-PLUMBER (Protocol for the Analysis of Land Surface Models Benchmarking Evaluation Project) was conducted (Lipson et al., 2024), summarising the most recent developments of the ULSMs based on 30 selected representative examples. It compared the capability of the models, including building complexity, hydrological complexity, and behavioral complexity. The study concluded that over the past decade, advancements in urban land surface models have led to a more accurate representation of urban morphology with good input data, coupled with focussed efforts on vegetation and hydrological processes. Recently, Jongen et al. (2024) evaluated the water balance in 19 ULSMs from the Urban-PLUMBER project (Lipson et al., 2024), concluding that the water balance appears unclosed in 43% of the model runs. The interactions between the urban water cycle and the energy budget are therefore not comprehensively captured in the existing ULSMs.

Hydrological models have a long development history and have become increasingly important tools for managing water resources. Regarding hydrological processes in urban areas, there is no universally accepted definition of the urban water cycle. However, many texts have previously agreed on dividing the system into two main networks: modified natural pathways and supply-sewerage pathways (Lerner, 2002; Dwarakish and Ganasri, 2015). This division, however, has become less strict in recent years as the increased inclusion of urban greening and blue-green infrastructure in the urban environment (Fletcher et al., 2024) has led to urban irrigation using drinking water (e.g., Guo et al. (2021)), hence linking the modified natural and supply-sewerage components. Thus, *urban hydrological models* (UHMs) must consider natural hydrological processes under artificial modifications and hydraulic processes (Urich and Rauch, 2014). The hydrological modelling of urban areas is highly challenging as urban areas are strongly heterogeneous and exhibit very specific hydrological processes (Ichiba et al., 2018). Salvadore et al. (2015) reviewed 43 hydrological models applied to urbanized areas, noting that the main differentiation can be found in spatial and temporal scales of models and various functions as opposed to modelling schemes or approaches. One of the notable conclusions drawn by Salvadore et al. (2015) was that although all the included models consider precipitation and evaporation analysis, a few consider energy balance variables.

Historically, the urban thermal and hydrological environments have been treated as distinct fields, which were studied independently. As our understanding deepens and the urgency to address future climate disasters grows, it becomes clear that hydrological extremes—such as floods, droughts, severe urban thermal environments, and more frequent heat waves can not be isolated events. These extreme events can occur simultaneously or successively, creating compound events (Wehner, 2023; Zscheischler et al., 2020). Evaluating the impact of these compound events is crucial, as it provides a more comprehensive understanding than considering each event in isolation. Research indicates that heat waves following floods have caused more

severe damage, as observed in Japan, where the 2018 floods were followed by a heat wave with temperatures exceeding 39°C (Wang et al., 2019), and South Asia, where Pakistan faced compound damage to infrastructure and agriculture (Lau and Kim, 2012). Research also pointed out that summer floods in the central United States are often preceded by a heat stress event (Zhang and Villarini, 2020). Cities like Miami and New Orleans in the United States, as well as some South European cities, faced severe droughts exacerbated by the urban heat island effect, leading to extreme heat conditions and increased strain on

water resources (Aboulnaga et al., 2024). To enhance urban safety and livability, various mitigation measures have been proposed and implemented. Among these, urban blue-green spaces are particularly effective, as they not only mitigate the urban heat island effect but also reduce surface runoff, benefiting urban hydrological and the thermal environment (Gunawardena et al., 2017; Li et al., 2019). Wang (2021) proposed a mathematical framework for unified evaluation of compound impacts emphasizing that compound urban climate mitigation should be evaluated in a comprehensive way. The compound extreme

events and mitigation mechanisms highlight the crucial role of the interaction between the water cycle and energy balance. For instance, urban water conservancy projects influence the moisture content of surface and near-surface soils; integrating urban drainage facilities into landscapes with water features may cool down urban temperature (Ferdowsi et al., 2024). Additionally, the interaction between the temperature of urban water bodies and the near-surface, and urban thermal conditions influence the water body ecosystem and water quality (Chen et al., 2023; Brans et al., 2018; Grey et al., 2023). These processes underscore

the intricate link between urban meteorology and hydrology (Oke, 1982; Fletcher et al., 2013).

Current process-based simulation tools, such as ULSMs and UHMs, have evolved independently, each driven by distinct considerations and applications. This divergence has resulted in limited overlap and interaction between the two. Consequently, accurately simulating urban hydrological and near-surface meteorological conditions, collectively referred to as hydro-meteorological conditions, simultaneously remains challenging. This makes it difficult to assess the risk of extreme events,

particularly compound events, and to develop diverse strategies to address future climate change. Until now, significant knowledge gaps in hydro-meteorological dynamics persist within the intricate interplay of urban morphology, human activities, and the reciprocal effects of heat fluxes, radiation, and hydrological processes (Grimmond et al., 2020). Previous literature reviews have conducted comparative analyses on one of these two groups of models, but rarely have they compared the two groups of models. It is urgent to point out the future development direction of physical models to meet the needs of hydro-meteorological

prediction and assessment to better cope with the occurrence of extreme event, as well as the feasibility assessment of efficient response measures in densely populated urban areas.

Triggered by the increasing demands of studying hydro-meteorology in urban areas, the current study's main objective is to summarise, assess, and formulate recommendations to effectively develop numerical simulation tools that can represent the key urban hydro-meteorological processes. To bring the urban hydrological and meteorological fields together and assess the future

of hydro-meteorological models, an inventory of the current state-of-the-art of meteorological and hydrological models was made. Fig. 1 shows the overall structure of the current review. First, both ULSMs and UHMs are selected based on previous literature reviews, with extra attention focusing on the hydro-meteorological interface; second, we summarise the similarities and differences between the selected ULSMs and UHMs, respectively; third, we assess models' ability to simulate the main urban hydro-meteorological processes. Based on the results, we identify gaps between the ULSMs and UHMs and weaknesses

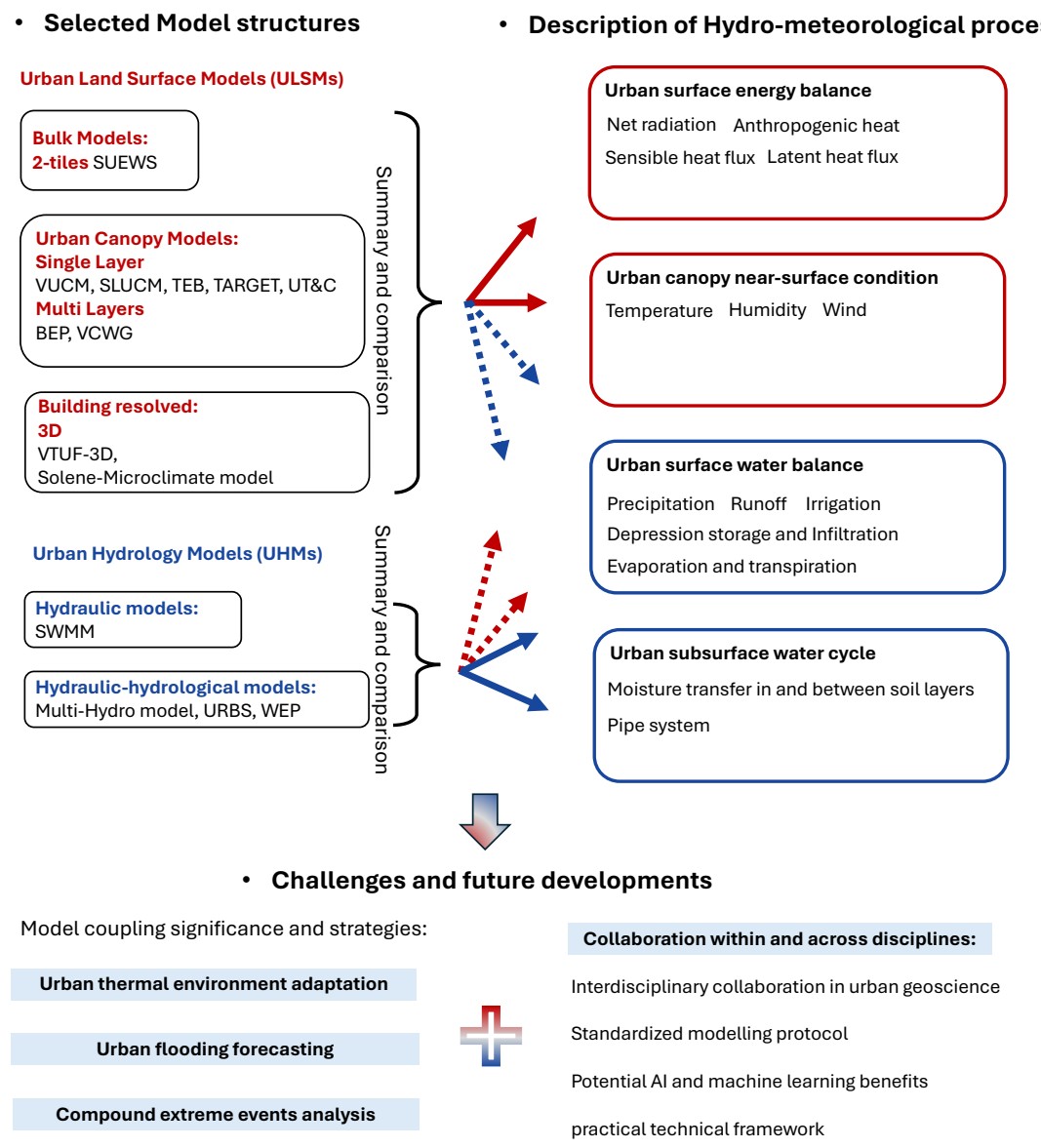

**Figure 1.** The overall schematic structure of the review.

and advantages of the models, resulting in the identification of potential coupling opportunities between ULSMs and UHMs that can offer insights for future model development in urban hydro-meteorology.

## 2 Selection of numerical simulation models

To bring the urban hydrological and meteorological fields together and assess the future of hydro-meteorological models, an inventory of the current state of the art of meteorological and hydrological models was made. Both research-based and applied models were considered. The models considered are shown in Table 1 and Table 2, which lists the long and short names of the included models with the main literature on model development. The listed literature encompasses not only the initial development of the model but also subsequent enhancements by various researchers. However, it excludes studies where the model was applied. The latest version of the model does not ensure the inclusion of all developed physical processes. We consider all added physical processes as part of the model's existing simulation capabilities.

Depending on the complexity of the model (Lipson et al., 2024), the ULSMs can be divided into:

(1) Bulk Models: These models treat the urban area as effectively a kind of bare soil, albeit with modifications. Bulk models parameterise heat capacities, thermal conductivities, surface albedo, roughness length, and moisture availability. The bulk models can be separated by how many urban facets are considered. The model is called the one-tile if it only has one urban facet, treating the urban area as a whole. There are also two-tile models with two urban facets: the roof and the ground.

(2) Urban Canopy Models (UCMs): UCMs average the characteristics of buildings and assume an infinitely long street canyon. The urban canyon is the area beneath the rooftop and is flanked by buildings on both sides. Consequently, canopy models primarily operate in two dimensions (2D). The models have at least three urban facets: wall(s), roof, and ground. Depending on the vertical resolution the model resolves in the urban canyon, the model can be either single-layer or multilayer. The single-layer models treat the urban canyon as a homogeneous area. However, the multilayer models solve the vertical profiles of the canopy-layer flows and momentum transport.

(3) Building-Resolving Models: These models normally employ computational fluid dynamics (CFD) to accurately simulate thermal and airflow conditions with 3D information of the buildings and heterogeneous urban landscape. Thus, all the urban facets are represented.

Referring to the previous review works, several ULSMs were included in this study for the later analysis of included physical processes. The current study includes ten ULSMs, including a two-tile bulk model, seven urban canopy models, and two building-resolving models. Five out of ten ULSMs have been coupled with mesoscale land surface schemes. Here, we only consider the Weather Research and Forecast model (WRF) as the mesoscale model example. Not every ULSM is included in the summary of our analysis. We only selected some representative, widely used, and newly developed models based on our focus. Since we aim to focus on the interface of hydro-meteorology in the urban environment, the representation of vegetation, water surface, and the urban water balance in the ULSMs were given extra attention during the selection of the models and model versions. Studies using CFD models to explore microscale urban climate, energy balance, and wind environment are common, but rarely focus on water balance processes (Lipson et al., 2024). For example, PALM has recently been intensively developed with different modules to suit the urban layout (PALM-4U), but it has not yet been widely evaluated and applied

**Table 1.** List of the included ULSMs

| Short name | Long name | Reference |
|---|---|---|
| SUEWS (WRF-SUEWS) | Surface Urban Energy and Water Balance Scheme | Grimmond et al. (1986); Grimmond and Oke (1991); Järvi et al. (2011, 2014); Ward et al. (2016); Järvi et al. (2017); Omidvar et al. (2022); Sun et al. (2023) |
| VUCM (WRF-VUCM) | Vegetated Urban Canopy Model | Lee and Park (2008); Lee (2011); Lee et al. (2016) |
| ASLUM (WRF-SLUCM) | Arizona State University Single-Layer Urban Canopy Model | Kusaka et al. (2001); Wang et al. (2013); Sun et al. (2013); Vahmani and Hogue (2014) ; Yang et al. (2015); Ryu et al. (2016); Wang et al. (2016); Upreti et al. (2017); Wang et al. (2021a) |
| TEB (WRF-TEB) | Town Energy Budget | Masson (2000); Hamdi et al. (2011); Stavropulos-Laffaille et al. (2018); Redon et al. (2020); Meyer et al. (2020); Stavropulos-Laffaille et al. (2021); Colas et al. (2024) |
| TARGET | The Air-temperature Response to Green/Blue Infrastructure Evaluation Tool | Broadbent et al. (2019) |
| UT&C | Urban Tethys-Chloris | Meili et al. (2020, 2025)                              ; |
| VCWG | Vertical City Weather Generator | Moradi et al. (2021, 2022) |
| BEP (WRF-BEP) | Building Effect Parameterisation | Martilli et al. (2002); Krayenhoff et al. (2014); Krayenhoff et al. (2020); Yu et al. (2022) |
| VTUF-3D | Vegetated Temperatures Of Urban Facets | Krayenhoff and Voogt (2007); Duursma and Medlyn (2012); Nice et al. (2018) |
| Solene-Microclimat model | Solene-Microclimat model | Robitu et al. (2006); Musy et al. (2021); Robineau et al. (2022) |

**Table 2.** List of the included UHMs

| Short name | Long name | Reference |
|---|---|---|
| SWMM | Storm Water Management Model | Gironás et al. (2010); Rossman and Huber (2016); Rossman (2017); Rossman and Simon (2022) |
| Multi-Hydro model | Multi-Hydro model | Ichiba et al. (2018) |
| URBS | Urban Runoff Branching Structure | Rodriguez et al. (2003); Berthier et al. (2004); Rodriguez et al. (2008); Pophillat et al. (2021) |
| WEP | Water and Energy transfer Processes | Jia et al. (2001); Li et al. (2017) |

(Resler et al., 2017; Gehrke et al., 2021). The development of the ENVI-met is not well documented since it is a commercial software package. Neither PALM nor ENVI-met is included in the current review (Maronga et al., 2020; Tsoka et al., 2018). It is also worth mentioning that the current review does not assess the performance of the individual models; for this, readers are referred to Lipson et al. (2024) and Jongen et al. (2024).

Hydrological models are typically categorised by their spatial resolution into lumped, semi-distributed or distributed models (Khakbaz et al., 2012). However, this review focuses on the processes modelled rather than their spatio-temporal resolution. For this reason, the UHMs included in the current review were separated into:

(1) Hydraulic models: in a narrow sense, the urban hydrological models refer to the type of models whose main function is to quantify the inflow into stormwater drainage (and sewers) derived from rainfall-runoff processes. This is often coupled with a hydraulic model to simulate the routing inside the drainage conduits. A wide range of commercial and open-source software packages are available for this purpose. One good example is the Storm Water Management Model (SWMM) (Gironás et al., 2010). The description of typical natural hydrological processes is simplified for this group of models (Salvadore et al., 2015).

(2) Hydraulic-hydrological models: this group of models with the main focus on the general hydrological simulations, usually used for evaluating the impact of urbanisation, have limited treatment of artificial pathways (sewers, stormwater systems, water supply).

The selection of UHMs also focuses on how they contribute to the interface of climatology and hydrology simulations. As mentioned by a previous review study (Salvadore et al., 2015), most UHMs do not include the surface energy balance and climatological process interactions. Only the weather conditions during the simulation period, such as precipitation and temperature, are general input for the models. The hydraulic models selected in the current study are mainly based on SWMM. There exist other simulation tools used for urban water management, such as MIKE URBAN, developed and managed by the Danish Hydraulic Institute (DHI, 2024), InfoWorks ICM, owned by AutoDesk (Wu et al., 2021a) and the Delft3D model (Hasan Tanim and Goharian, 2021). However, many of them are not open source and focus specifically on wastewater treatment, sewer system design, and coastal urban floods. Most of the hydro-meteorological interfaces in the models share major

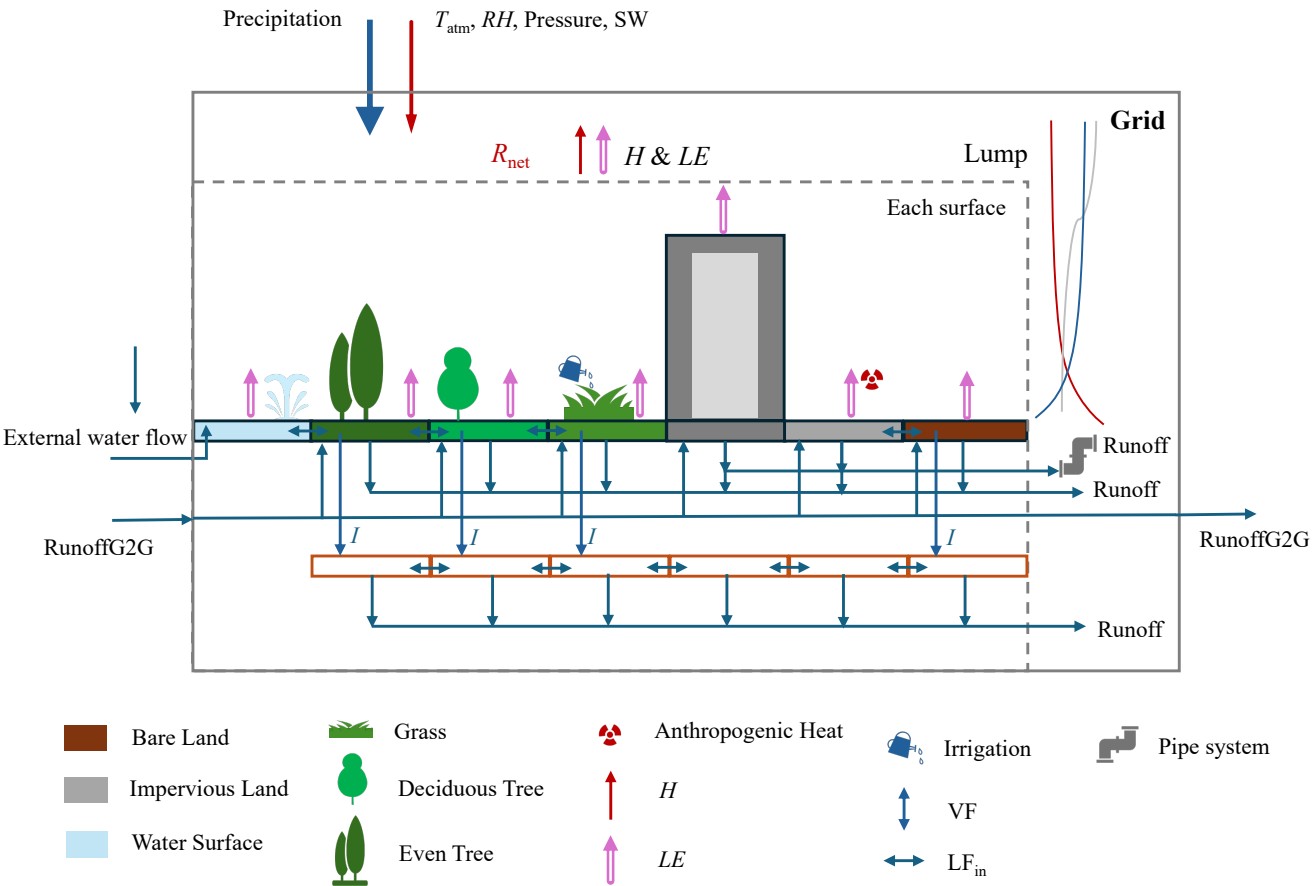

**Figure 2.** Schematic structures of SUEWS and the included elements and main processes calculated in the model are represented. Redrawn after the Figure 1 in Järvi et al. (2014). $H$ is the sensible heat flux, $LE$ is the latent heat flux, VF presents the vertical flow and $LF_{in}$ presents the lateral flow within grid cesll. The meaning of the abbreviations in the figure can be found in the List of Symbols and Acronyms.

components similar to SWMM. Thus, our selection can be considered representative of the majority of urban hydraulic mod-
els. Three hydraulic-hydrological models that mainly consider the general urban hydrological simulations are selected in the
current study for later analysis. Each model has a different spatial and process complexity level, which will be analysed later.

## 3 Urban land surface models (ULSMs)

### 3.1 Bulk models

The only bulk model selected, the Surface Urban Energy and Water Balance Scheme (SUEWS), is a two-tile model, which
was first released by Järvi et al. (2011). The two-tile model resolves two urban surface facets (roof and street canyon) with

different thermal and radiative properties. The two-tile model only resolves two surface energy balances. The primary objective of SUEWS is to be capable of simulating energy and water fluxes and getting the output at an hourly level temporal resolution (therefore requiring relatively low computational complexity) using minimal input data, which can be obtained from standard meteorological stations or an urban mesoscale model. The model is derived from the model developed by Grimmond et al. (1986), who initially introduced a water balance model for urban areas, considering actual evaporation rates at the neighbourhood or local scale (ranging from 100 meters to 1 kilometer). Additionally, Grimmond et al. (1991) employed an evaporation-interception approach to calculate hourly fluxes. Järvi et al. (2014) further developed the model for cold climates, including snow-related processes. Diagnosis of air temperature, humidity, and wind speed within the roughness sub-layer was updated by Theeuwes et al. (2019); Tang et al. (2021).

Figure 2 shows the schematic SUEWS and the included physical processes. The model takes the precipitation, first layer atmosphere air temperature and pressure, and shortwave radiation as inputs. Seven land use types are included in SUEWS, with a single soil layer. The surface temperature ($T_s$) and net radiation ($R_{net}$) are solved separately for each land use type and the building roof, but sensible heat ($H$) and latent heat ($LE$) fluxes are solved for the whole grid with the lumped (gridded average) result. There is no interaction between roof and land surface tiles. The model has the ability to compute the grid-average vertical profile of the near-surface air temperature, humidity, and wind. The hydrological cycle is included to some extent, with infiltration ($I$) and lateral water flow among different types of land surface ($LF_{in}$). The $LF_{in}$ is simulated by the following steps: first, each surface calculates its drainage based on the water balance equations; second, the water distribution matrix, which specifies how water is redistributed among different surface types, will be used for the surface lateral flow process (Grimmond et al., 1986; Järvi et al., 2011). For example, water drained from a paved surface might be partially redirected to nearby grass or soil surfaces. Third, when the infiltration capacity is exceeded, excess water becomes runoff. The model calculates potential runoff for each surface type and redistributes it according to the water distribution matrix. Although grid-to-grid runoff (RunoffG2G) is included as an external fixed term in the water balance, the dynamic interaction between grids is not included.

### 3.2   Urban canopy models

UCMs are ULSMs that resolve the energy balance for the roof, wall, and ground surface separately. The current assessment includes seven ULSMs belonging to the canopy scheme, making it the most abundant ULSM type assessed in this review. Masson (2000) developed a single-layer urban canopy model called the Town Energy Budget (TEB) scheme. Similarly, Kusaka et al. (2001) introduced the Single-Layer Urban Canopy Model, which is integrated into the Weather Research and Forecasting model (WRF-SLUCM). Unlike TEB, which uses analytical formulas for in-canyon view factors, WRF-SLUCM employs a distinct radiation parameterization scheme by discretizing canyon facets and calculating radiation for individual grid cells. Although Kusaka's scheme may be less efficient for simple rectangular canyons with walls, roads, and short vegetation, it has the advantage of accounting for radiative exchanges involving roughness elements such as trees, blocks, and vehicles within street canyons. The ASLUM was developed following Kusaka's SLUCM and has been updated to the present (Wang et al., 2013). A multi-layer UCM was developed by Martilli et al. (2002) and is named the Building effect parameterisation

(BEP). The primary objectives of UCMs are to focus on energy and momentum exchange between an urban surface and the atmosphere. Thus, UCMs are generally good at reproducing surface temperatures and sensible heat budgets. However, they are inevitably inadequate in capturing the dynamics of the urban water budget and the latent heat due to the lack of realistic urban hydrological processes incorporated.

Motivated by various research requirements, including investigations into the cooling influence of vegetation in urban street
canyons and the effects of urban water circulation on the thermal environment, UCMs have progressively incorporated considerations related to urban vegetation and hydrological processes (Grimmond et al., 2010). Lee and Park (2008) first introduced the Vegetated Urban Canopy Model (VUCM), which comprehensively accounts for the impact of vegetation on wind speed, radiative energy partitioning, soil processes, and vegetation energy budgets within an urban canyon. Wang et al. (2013) proposed a surface exchange scheme coupling the transport of energy and water in urban canopies for the ASLUM. Sun et al. (2013)
incorporated green roof-related processes based on the version of Wang et al. (2013) ASLUM. The parametrisation scheme for urban trees also gradually included the shading effect and evaporation in the stand-alone ASLUM (Ryu et al., 2016; Wang et al., 2016, 2021a). Urban trees have also been included in BEP but do not include the hydrological module (Krayenhoff et al., 2014, 2020). Lemonsu et al. (2007) introduced both the rainfall interception capacities of built-up surfaces and integrated water infiltration through artificial surfaces into the TEB, and later, the vegetation scheme was also implemented for direct
interaction between the urban landscape with vegetation and built-up area (Lemonsu et al., 2012; Redon et al., 2017, 2020). A complete hydrological scheme for urban areas was developed by integrating the subsoil under built-up surfaces and hydrological soil–surface interactions into the vegetation version of TEB (Stavropulos-Laffaille et al., 2018, 2021).

Recently, newly developed models have focussed more on assessing the impact of blue and/or green spaces on urban microclimate. Broadbent et al. (2019) presented a new model called The Air-temperature Response to Green/Blue Infrastructure
Evaluation Tool (TARGET). As mentioned by the developers, TARGET is a simple simulation tool that calculates surface temperature and street-level (below roof height) air temperature in urban areas. It has been claimed that it is designed to make quick and accurate assessments of cooling impacts on green-blue spaces with minimal input data requirements. To further assess the impact of vegetation on urban climate and hydrology, Meili et al. (2020) presented an urban eco-hydrological model, Urban Tethys-Chloris (UT&C). The development of UT&C combines components of the eco-hydrological model Tethys-Chloris
(T&C) and ASLUM, thus including more detailed hydroclimatic processes. Besides, UT&C has the capability to consider physiological and biophysical properties of vegetation, and thus being able to consider different vegetation types, at least in principle, which was not the case in any of the other models except partially VTUF-3D (Meili et al., 2020, 2025). Moradi et al. (2021) developed a model called the Vertical City Weather Generator (VCWG), and the inclusion of hydrological processes is based on UT&C (Moradi et al., 2022).

Figure 3 presents the integration of all the UCMs that share a similar model structure but include different levels of physical complexity. Figure 3 includes all the physical processes that can be simulated by UCMs according to development literature. It is obvious that, unlike SUEWS, urban landscapes are included within urban canyons in UCMs. The UCMs take precipitation, air temperature, wind speed and specific humidity of the first layer of the atmosphere, as well as downward shortwave and longwave radiation and pressure as inputs. Building roofs directly interact with the first layer of the atmosphere in terms of

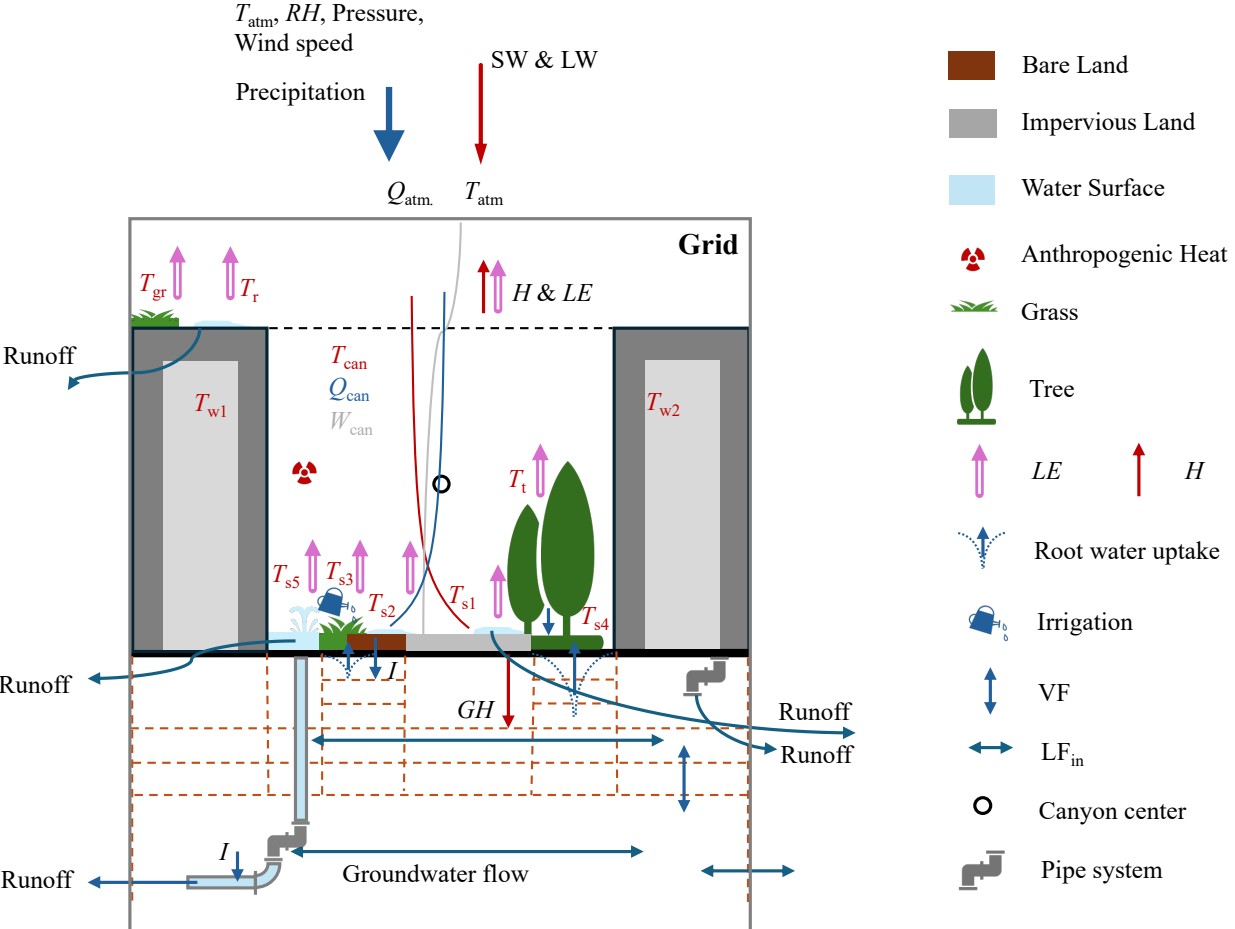

**Figure 3.** Schematic structure of UCMs and the included elements and main processes calculated in the model are represented. The meaning of the abbreviations in the figure can be found in the List of Symbols and Acronyms.

energy and humidity. The other surfaces, including the surface waters, vegetated land, bare land, building walls, and trees, directly interact with the urban canyon air. The exchanges of energy, water, and momentum take place between the urban canyon and the first layer of the atmosphere. The canyon-average air temperature, humidity, and wind can be solved, but the vertical profile of each simulation grid can only be solved in some of the UCMs. The soil layer can be single or multiple. The representations of vegetation and water surface are only included in some UCMs. The water balance, including infiltration, vegetation root uptake, urban pipe system inflow, lateral water flow between different land surfaces, and groundwater flow, can also be included based on the extent of development of each UCM. The canyon concept has been intensively adopted while developing simulation tools for the urban climate. Different UCMs have been developed by referring to each other but, meanwhile, vary due to the specific focus and developers' interests.

### 3.3 Building resolving models

3D-Models seldomly resolve the subsurface conditions and lack representation for the hydrological cycle, which is also why only two models are included in this study. The current analysis includes two building resolving models: the Vegetated Temperatures Of Urban Facets (VTUF-3D) model (Nice et al., 2018) and the Solene-Microclimat model (Musy et al., 2015).

VTUF-3D is the vegetated version of the Temperatures Of Urban Facets (TUF-3D) model (Krayenhoff and Voogt, 2007). The TUF-3D model was integrated with a model called MAESPA, which is the combination of the MAESTRO and SPA (Soil-plant-atmosphere) models, enabling the inclusion of any vegetation type in any configuration within a modelling domain to consider vegetation's structural and physiological processes. VTUF-3D uses cube-shaped structures (as TUF-3D uses to represent buildings) to represent vegetation. The structure of this coupled model can be seen in Figure 1 in Nice et al. (2018). The purpose of developing the VTUF-3D is to estimate accurate latent energy fluxes and the soil-vegetation-atmosphere pathway critical for vegetation and water-sensitive urban design assessments within the urban canyon, which is also the major reason we include this model in our analysis. However, the water balance is missing in the TUF-3D model (Krayenhoff and Voogt, 2007). The water balance is included only for the vegetation as mentioned in Duursma and Medlyn (2012). The flowchart of the water balance component for vegetation can be found in Figure 2 in Duursma and Medlyn (2012).

The Solene-Microclimat model was first developed for outdoor comfort assessment and then built the connection between building energy and indoor thermal conditions (Musy et al., 2015, 2021). The model is based on a 3D-representation of urban geometry, and the simulated scenes are modeled by the meshing of the 3D-developed surface with rectangular or triangular elements. It is a coupled model with three main parts: radiation model, thermal model, and airflow model. The radiation-thermal model is called the Solene model. The Solene-Microclimat model is completed by coupling the CFD model to simulate the airflow. The coupling was first performed by Vinet using the N3S CFD code, which has been replaced by FLUENT and later by Code Saturne (Robitu et al., 2006). More details can be found in Musy et al. (2021). Trees are considered a surface in the radiative model (the tree crown envelope) and a volume in the CFD model. Green roofs and walls, as well as lawns, are included in the model. A thermal model developed by Robitu et al. (2004) considers the radiation absorbed, transmitted, and reflected at the water surface. The model can also simulate the behavior of watered surfaces. The Modelling of Actual Runoff Infiltration and Evapotranspiration (MARIE), based on a distributed hydrological model, was later coupled with the Solene-Microclimat

**Table 3.** Information of included models. The information is based on the literature listed in Table 1 and Table 2 and also on Salvadore et al. (2015). The column of Temporal Resolution is marked by *, indicating the time scale at which model output is sampled.

| Shortname | Type | Simulation Unit | Spatial Resolution | Temporal Resolution* |
|---|---|---|---|---|
| SUEWS (WRF-SUEWS) | Bulk (2-Tile) | Grid cell | 100 m - 5 km | min - hr |
| VUCM (WRF-VUCM) | Canopy-Single layer | Grid cell | 100 m - 5 km | min - hr |
| ASLUM (WRF-SLUCM) | Canopy-Single layer | Grid cell | 100 m - 5 km | min - hr |
| TEB (WRF-TEB) | Canopy-Single layer | Grid cell | 100 m - 5 km | min - hr |
| TARGET | Canopy-Single layer | Grid cell | 100 m - 1 km | min - hr |
| UT&C | Canopy-Single layer | Grid cell | 100 m - 1 km | min - hr |
| VCWG | Canopy-Multi layer | Grid cell | 100 m - 1 km | min - hr |
| BEP (WRF-BEP) | Canopy-Multi layer | Grid cell | 100 m - 5 km | min - hr |
| VTUF-3D | 3D Building resolved | Mesh cell | 1 m - 10 m | sec - hr |
| Solene-Microclimat model | 3D Building resolved | Mesh cell | 1 m - 10 m | sec - hr |
| SWMM | 1D Hydraulic | Sub-catchment | 100 m - | min |
| Multi-Hydro model | 1D Hydraulic-hydrological | Grid cell | 100 m - | min |
| URBS | 1D Hydraulic-hydrological | UHE | 10 m - 1 km | min |
| WEP | 2D Hydraulic-hydrological | Grid cell | 100 m - 5 km | min - hr |

urban climate model (Musy et al., 2021). One of the main reasons for including this model in the analysis is that it shows the potential capability of coupling hydrological and land surface models.

### 3.4 ULSMs summary and comparison

With three different types of model structures (bulk model, UCMs, and building resolved models), the selected models have a wide spatial resolution coverage. The top part of Table 3 summarises the simulation unit name and spatial resolutions of the included ULSMs. The simulation domain is the combination of at least one simulation unit. The models solve each unit separately in most cases. The spatial resolution of the models indicates the general support of the simulation unit. Both bulk models and UCMs are 2D ULSMs. The standalone 2D ULSMs can be applied to a neighbourhood scale, sometimes called district scale, with a simulation resolution of 100 m - 1 kilometer (Figure 4a and b), called standalone mode hereafter. The 2D-ULSMs can be coupled with regional and global climate models, with a simulation resolution of 1 to 5 kilometers (Figure 4c and d), called coupled mode. The 3D building resolved models are, in general, only applied at the neighbourhood scale and have resolutions of the order of one meter (Figure 4e). The temporal resolution refers to the time scale for sampling the output. Although the analysis of the model output is a rather arbitrary choice for users, the table shows the general timescale analysed. The temporal resolution of the 2D ULSMs is above half an hour and can reach the minimum level for the 3D ULSMs.

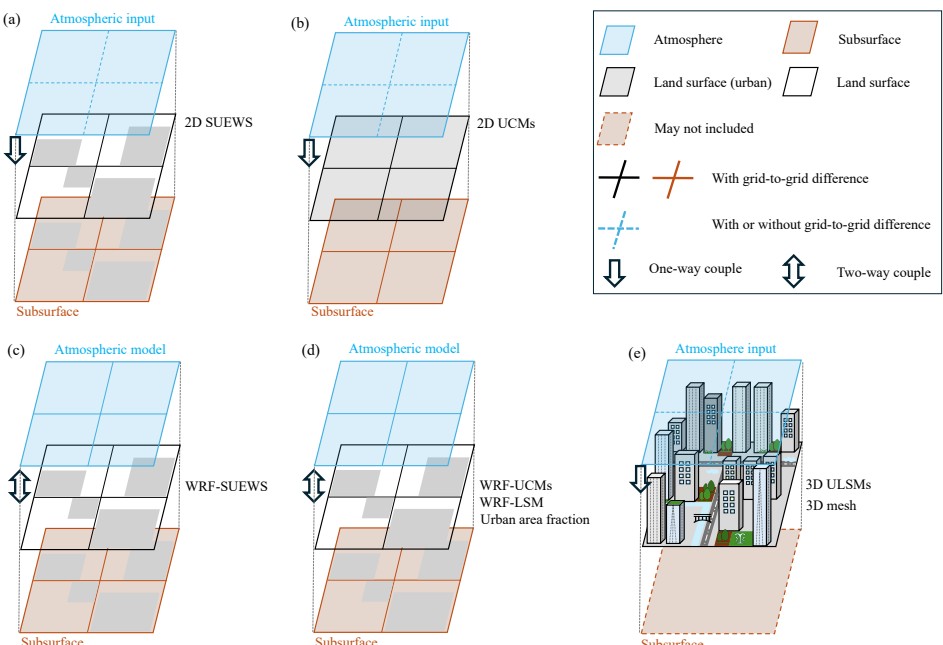

**Figure 4.** Conceptual figure of model structure and coupling condition for standalone ULSMs (a) and (b), coupled ULSMs (c) and (d), and building resolved ULSMs (e).

Figure 4a shows the structure of the selected standalone 2D bulk model (SUEWS). The model can simulate the impervious land area and the other six landscapes, as mentioned in Figure 2. Figure 4b shows the structure of the UCMs. It can be seen
that UCMs assume the whole study area is within the urban canyon. Both standalone model SUEWS and UCMs focus on the study area in one or several independent simulations. If the study area is separated by different landscapes (four grid cells used as an example here), the standalone mode will solve each grid cell independently, which means: 1) the landscape and urban morphology are assumed to be homogenous within each grid cell; 2) there is no interaction between grid cells. The atmospheric condition for each single grid cell can differ for the standalone mode depending on the user setting and data
availability, represented by the dashed blue lines in the top grid cell of Figures 4a and b. It is worth mentioning that the 'grid cell' here in Figure 4a and b does not need to be a strict square shape. Once we assume a certain urban neighbourhood shares the same morphology, an individual standalone mode simulation can be set up.

Figures 4c and d show the structures of the coupled mode in climate models, which simulate the entire study area in one go. These models use square grid cells of uniform size. SUEWS, similar to a complete land surface model, includes impervi-
ous land. Recently, WRF-SUEWS (Sun et al., 2023) was developed as a land surface scheme. The main difference between standalone and coupled SUEWS is the interaction of atmospheric conditions with the land surface. Coupled mode uses Urban Canopy Models (UCMs) for impervious areas, with the fraction set manually based on urbanization or land use data. The land surface model handles other landscapes like grass or bare land. Interactions between impervious and other land areas occur

indirectly through air exchange. Coupled mode simulations resolve grid-to-grid atmospheric differences but limit grid-to-grid interactions. Coupled mode 2D UCMs focus on impervious land, while standalone UCMs integrate various urban canyon processes (Ryu et al., 2016; Wang, 2014; Lemonsu et al., 2012). Thus, standalone models are usually more advanced and comprehensive for the urban process (Krayenhoff et al., 2020; Meyer et al., 2020).

Building resolving models (Fig. 4e) solve the simulation area with the 3D mesh concept with resolutions down to 1 meter. 3D models can be scaled up to mesoscale models, providing detailed three-dimensional meteorological variables. However, due to their computational demands and substantial initial and boundary conditions, building resolving models are predominantly applied at micro-scale levels (less than 100 meters) and less frequently in regional or global climate simulations. The upper atmosphere condition and meteorological information are usually one way forcing the simulations.

The trend in developing 2D models is towards higher complexity than bulk models. Unlike bulk models, UCMs incorporate the urban canopy concept, allowing for the analysis of more intricate influences within urban areas and street canyons, such as the effects of street trees and urban green spaces. Consequently, UCM models are more widely utilised for studying regional impacts at the hundred-meter scale, and high-resolution CFD models are primarily used for detailed research at the meter scale.

## 4 Urban hydrological models (UHMs)

### 4.1 Hydraulic models

In the context of stormwater drainage and sewer flow simulations, we have opted to present SWMM, one of the most widely employed urban watershed hydrological models (Niazi et al., 2017) as the representative model containing overland hydrology and subsurface hydraulics. Our choice of SWMM serves as an illustrative representation for this type of model and was chosen due to its open-source nature and therefore interpretability of the model structure. The schematic figure inside the model is shown in Figure 5. The atmospheric conditions, including air temperature and precipitation, are inputs to the model. There are three types of subareas within each sub-catchment: pervious land and impervious land with and without depression storage. Water can be transferred among the different subareas within a simulated unit though it is strictly zero-dimensional. Evaporation and transpiration can be solved either through the inclusion of meteorological parameters or are normally taken as input time series. The household wastewater, piped system (resolved in one-dimension), water treatment process, and low-impact development, which includes green vegetation, are also included in SWMM (Fig. 5). The subsurface is represented by a two-zone scheme: an unsaturated zone and a saturated zone. Building and urban morphology are not present in SWMM. The spatial resolution of the hydraulic model is largely dependent on model requirements and data availability. Usually, individual conduits are considered, with the surface characteristics lumped per manhole (e.g., Nedergaard Pedersen et al. (2021)), though the entire network can also be lumped into a single hydrological model within the SWMM environment (e.g., Farina et al. (2023)).

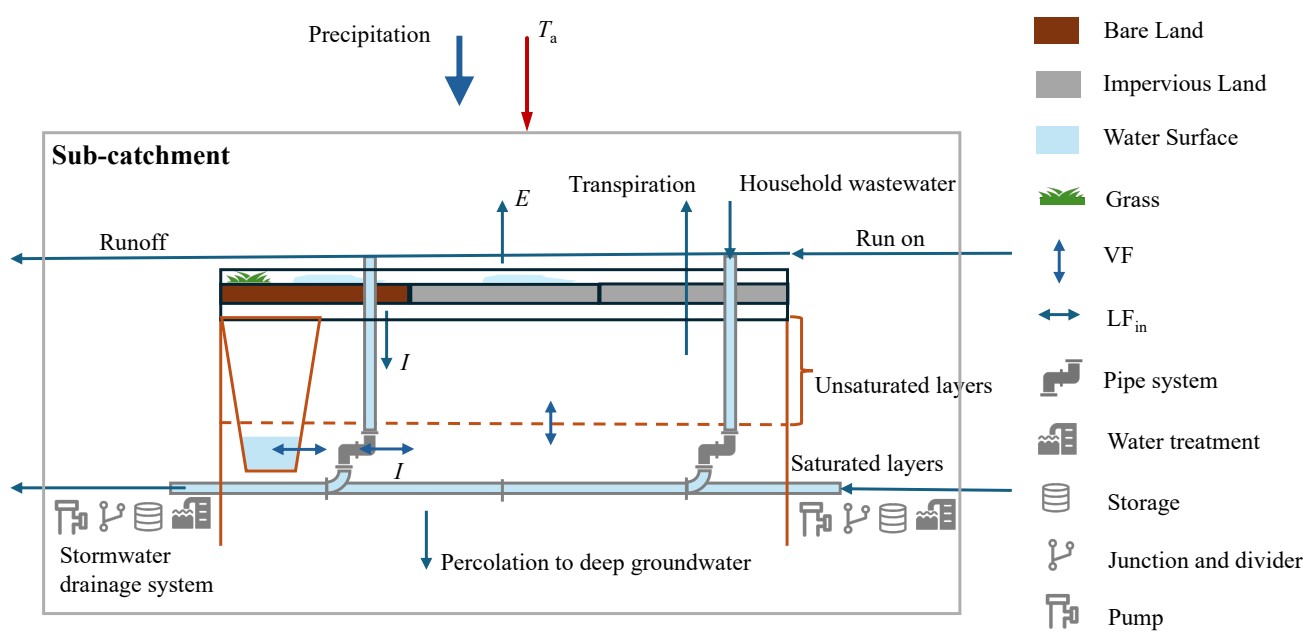

**Figure 5.** Schematic structure of SWMM and the included elements and main processes calculated in the model are represented (drawn by referring to Figure 3-1 and Chapter 3 in Rossman and Simon (2022)). The meaning of the abbreviations in the figure can be found in the List of Symbols and Acronyms.

## 4.2 Hydraulic-hydrological models

Rodriguez et al. (2008) developed a hydrological model, the Urban Runoff Branching Structure (URBS) model (see Figure 6 for the model structure). The atmospheric conditions, precipitation, and air temperature are the inputs. The geometry information of the urban catchment, including the houses, street sections, and vegetation; topography, street segments, storm sewers, and rivers are provided by a database (Urban Databank) (Rodriguez et al., 2003). URBS solves each simulation unit with three profiles for three land use types: natural soil, street, and house. The trees are represented 'above' the land surface with a depression storage

function for precipitation water. The subsurface of each land use type is the same, namely a two-zone scheme. Pophillat et al. (2021) implemented the groundwater section, including interactions between surface hydrology, groundwater, and underground structures in URBS. URBS can be coupled with the Solene-Microclimat urban climate model, a 3D building resolved ULSM mentioned in the above Section 3.3 (Robineau et al., 2022). The procedure for the coupling between the microclimate and hydrological models can be found in Figure 4 in Robineau et al. (2022). Although the two models are operated independently

and the data exchange is done manually, this coupling study shows the potential of simulating the urban hydro-meteorological interface (Robineau et al., 2022).

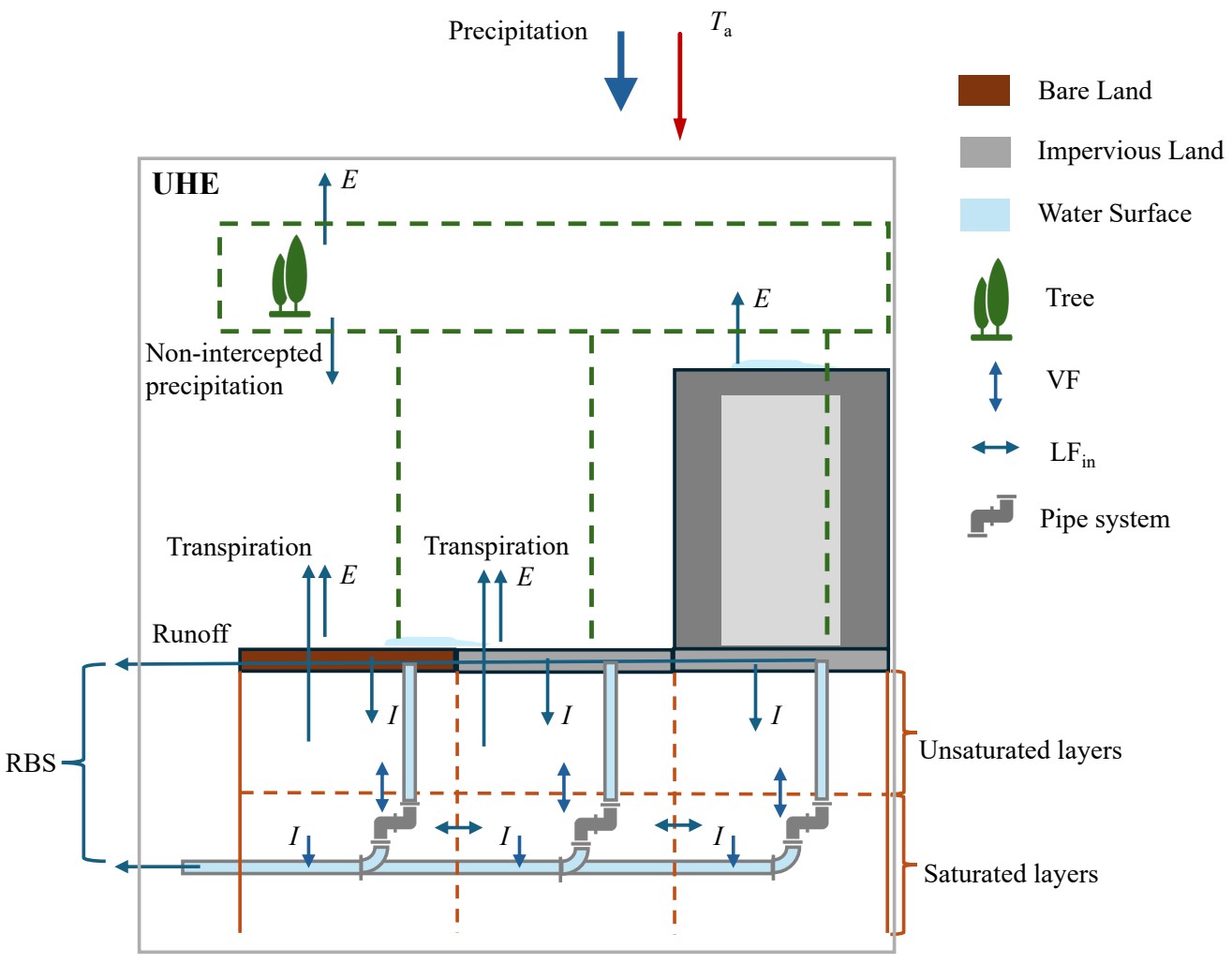

**Figure 6.** Schematic structure of URBS and the included elements and main processes calculated in the model are represented (redrawn after Figure 2 in Rodriguez et al. (2008)). The full name of the abbreviations in the figure can be found in the List of Symbols and Acronyms.

Another hydraulic-hydrological model is called Multi-Hydro (Ichiba et al., 2018). As the Multi-Hydro model is more like a coupled model with four modules(rainfall, surface, soil, and drainage), the schematic structure of Multi-Hydro is not shown here and can be found in Figure 1 in Ichiba et al. (2018). The four open-source software packages are integrated, each representing a portion of the water cycle in urban environments. As mentioned by Ichiba et al. (2018), seven parts are generally included: precipitation, interception and storage, infiltration, overland flow, sewer flow, infiltration into the subsurface zone, and sewer overflow.

The last included model, the water and energy transfer processes (WEP) model, is developed by Jia et al. (2001). Figure 7 displays the process included in WEP. WEP is the only hydrological model we selected that includes the energy balance. The precipitation and incoming radiation are the inputs for WEP. Three land use types are distinguished: water surface, pervious land (bare soil), and impervious land. Buildings and vegetation are included in WEP. Evaporation, together with latent heat flux, is solved. Root water uptake, infiltration, and water leakage are included. The unsaturated and saturated subsurface layers are represented in WEP. The model has four layers in the unsaturated zone. The saturated zone includes one unconfined aquifer, two confined aquifers, and in-between aquitard layers. However, it is worth mentioning that the pipe system is not included in WEP. This means that WEP cannot be used for urban drainage design and planning, the major objectives for SWMM. WEP is in that sense more similar to a hydrological model than a hydraulic model.

### 4.3 UHMs summary and comparison

The lower part of Table 3 lists the simulation unit and resolution information of the included UHMs. Most of the included models have a spatial resolution larger than 100 meters (Salvadore et al., 2015). Figure 8 shows the simulation structure of the included UHMs. All selected UHMs do not have two-way coupling with the atmosphere. The atmospheric inputs can come from weather stations, as well as regional weather simulations. Each UHM has a different simulation unit and unit linkage. In SWMM, each sub-catchment has an inlet node to the pipe system or channel. The simulation domain is subdivided into sub-catchments. Each sub-catchment is assumed to be an idealised rectangular area with uniform properties and a uniform slope and width (Figure 8a). URBS simulates the hydrological processes at the urban hydrological element (UHE) scale. An UHE encompasses a cadastral parcel and its corresponding adjacent street segments. It considers an urban catchment to be composed of a set of UHEs connected to the catchment outlet by means of a runoff branching structure (Figure 8b). WEP is a grid-based model. The grids are linked with a one-dimensional channel and then linked with the main river. The Multi-Hydro model is also grid-based, with the land use type assumed to be homogeneous within a grid. Thus, the model structure is similar to the one shown for WEP in Figure 8c.

Urban hydrological models continue to face significant trade-offs due to highly complex and heterogeneous systems in which natural and anthropogenic processes interact at various spatial and temporal scales. One of the shortcomings of hydraulic models like SWMM is that they tend to underrepresent land surface processes and land-atmosphere interactions. Although WEP can solve the near-surface atmospheric conditions, it lacks a hydraulic system, which indicates that WEP is geared towards larger study areas (Li et al., 2017). Overall, it can be concluded that UHMs are generally unable to solve near-surface meteorological conditions. Unlike ULSMs, current UHMs do not need detailed urban morphology as the input. On the other

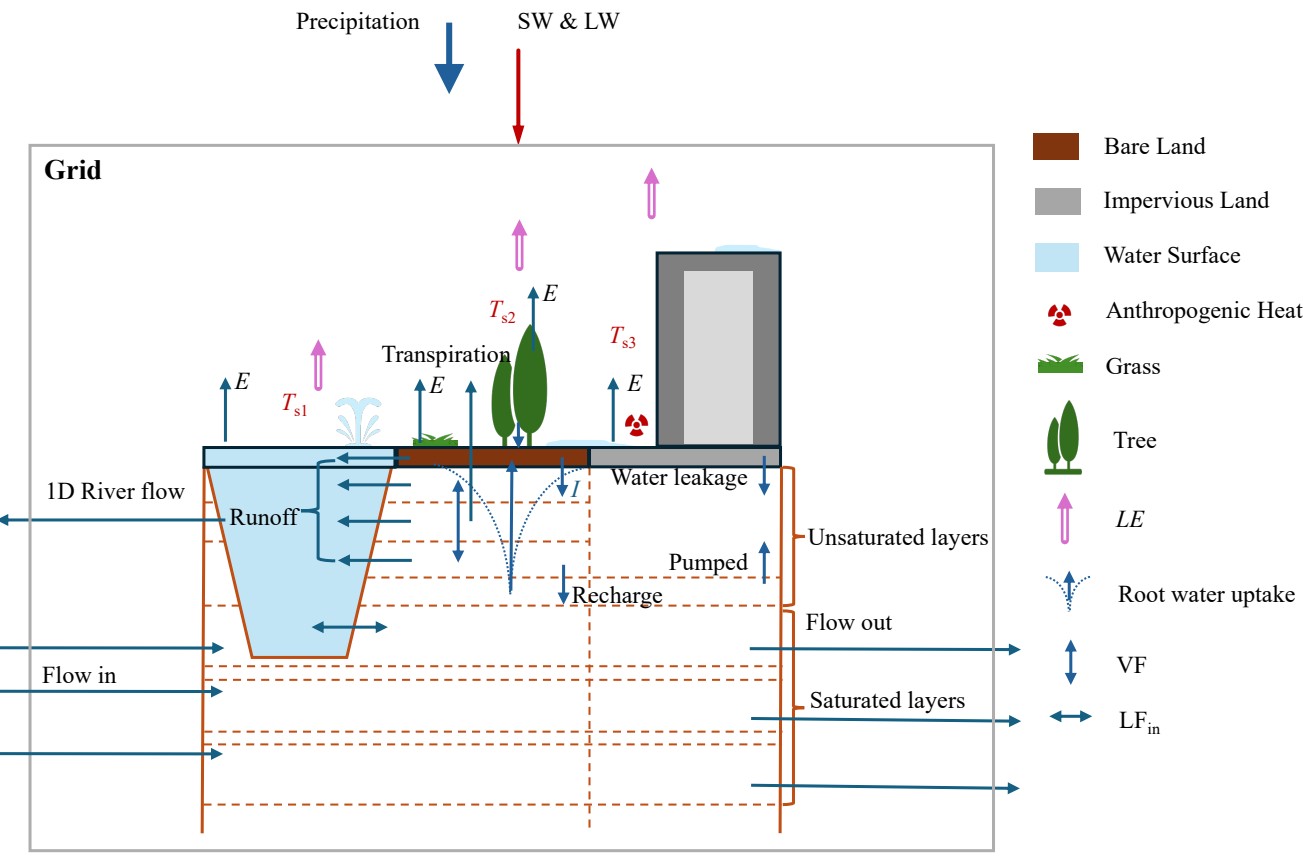

**Figure 7.** Schematic structure of WEP and the included elements and main processes calculated in the model are represented (redrawn after Figure 1 in Jia et al. (2001)). The meaning of the abbreviations in the figure can be found in the List of Symbols and Acronyms.

hand, UHMs are better equipped to simulate the grid-to-grid interaction for the hydraulic-hydrological processes on the surface and the sub-surface than ULSMs.

## 5  Simulations of urban hydro-meteorological processes

In the context of modelling urban environments, physical processes are governed by energy and water balance equations, which
are linked through the evaporation process. The various models analysed exhibit different levels of process complexity, based on their primary objectives. In this section, we analyse how the selected models implement the physical hydro-metereological processes and explore an alternative approach for incorporating hydro-meteorological interactions within the urban environment. Our examination begins with an exploration of the energy balance across different surfaces within the urban setting, followed by a summary of how the models simulate meteorological conditions for urban areas, especially for the near-surface.

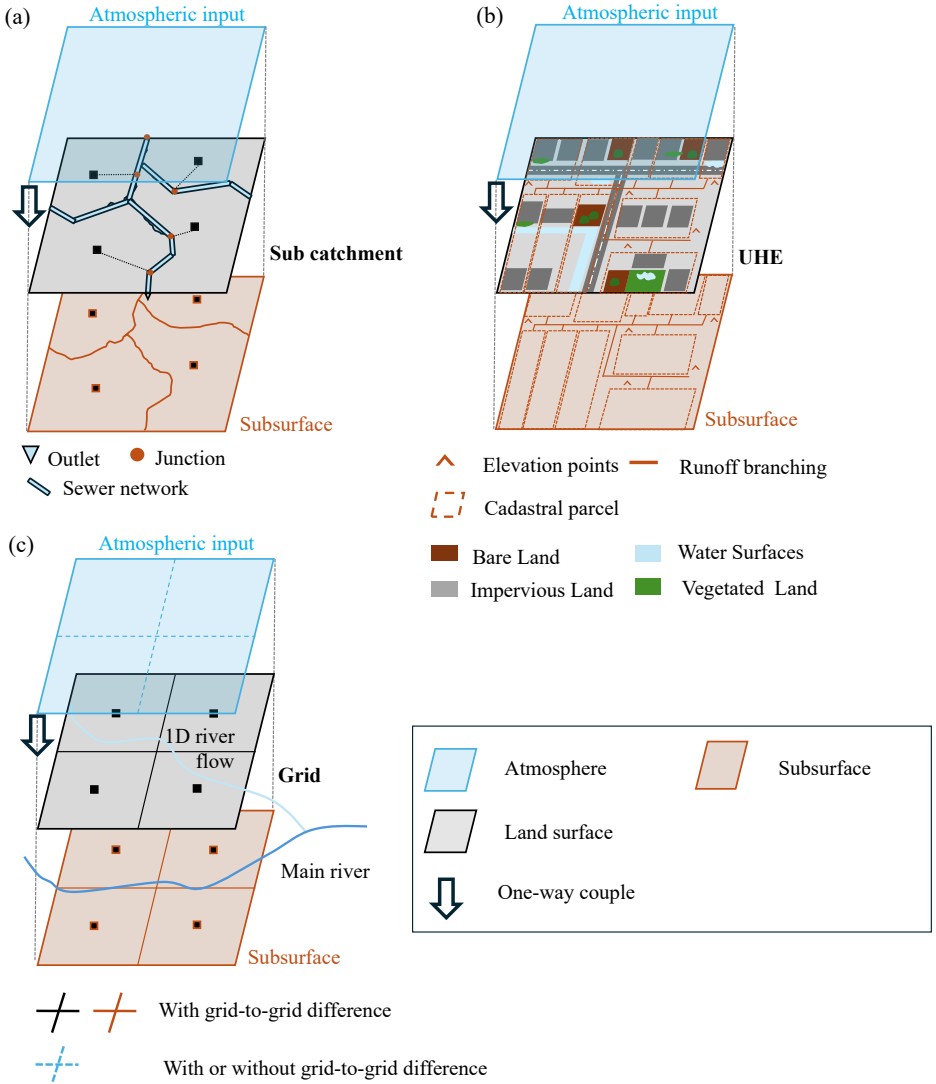

**Figure 8.** Conceptual figure of model structure and coupling condition for Urban Hydrological Models SWMM (a), URBS (b) and WEP (c).

Furthermore, we provide an overview of the hydrological processes affecting both surface and subsurface components. We separate the urban surfaces into impervious land, bare soil land, building facets (BudF), urban vegetation (VegL), and water surfaces (WatS). By summarising and comparing these aspects, we can identify gaps between the existing ULSMs and UHMs.

## 5.1 Surface energy balance

We systematically consider several key components from the energy balance equation: shortwave and longwave radiation (SW
and LW), sensible and latent heat flux ($H$ and $LE$), and anthropogenic heat release (AH). Table 4 displays the selected models' capability of simulating the surface energy balance. Note that Table 4 only includes the stand-alone versions of ULSMs. It can be seen that the surface energy balance constitutes a pivotal aspect of ULSMs. During the course of ULSM development, on the basis of simulating the energy balance of impermeable urban surfaces, the models added parameterisations of more complex and diverse urban surfaces, such as urban vegetation and bare soil land. The conductive and turbulent heat flux calculation is
based on temperature differences between the surfaces in the urban environment, air near the surface, and the upper atmosphere. It also depends on the aerodynamic resistance to the transport of heat, which is assumed to be equal to that of water vapor; thus, it is applied for calculating both sensible heat flux and latent heat flux. The latent heat calculation will be discussed in relation to evaporation and transpiration in Section 5.3.3. However, most ULSMs do not include the energy balance process for the urban surface water, like urban lakes (Table 4). Three exceptions are the SUEWS, TARGET, and Solene-Microclimat
models.

SUEWS simulates the water surface energy balance and the other land surfaces based on the net all-wave radiation parameterization scheme (NARP) and Local-Scale Urban Meteorological Parameterization Scheme (LUMPS) (Grimmond and Oke, 2002; Offerle et al., 2003). Unlike models based on the urban canyon concept, SUEWS calculates net radiation based on the input data: incoming shortwave radiation with relative humidity and air temperature or incoming long-wave radiation and cloud
cover can be used if available (Loridan et al., 2011). It is worth mentioning that SUEWS computes net radiation separately for individual surface types within a grid while solving the sensible, latent, and storage heat flux for the entire grid (Fig. 2). The sensible and latent heat fluxes are determined using the parameterised scheme proposed by Berggren et al. (2012), while the storage heat flux is solved using the objective hysteresis model developed by Grimmond et al. (1991).

As the name suggests, the main function of TARGET (The Air-temperature Response to Green/blue-infrastructure Evalua-
tion Tool) is to evaluate the impact of blue and green spaces on air temperature. Thus, including urban vegetation and water surfaces in the energy balance is particularly important (Bera et al., 2022). The LUMP model serves as the foundational component for the TARGET model (Grimmond and Oke, 2002). However, TARGET is specifically designed around the concept of urban canyons (Fig. 3). Within these canyons, various surface types are considered. Tree height is the same as roof height, and the areas beneath trees represent ground-level surfaces. The tree canopy is incorporated into the urban canyon representation
to account for the shading effects of trees. Additionally, sky view factors play a role in the net radiation calculations. Notably, the OHM–force–restore method employed in LUMP for solving surface temperature and storage heat flux across different surfaces is unreliable for reproducing water surface temperatures (Broadbent et al., 2019). TARGET employs a simple water body

**Table 4.** Selected model capabilities on simulating the surface energy balance process over different surfaces. 'Y' represents that the process is included, and 'N' for the opposite. Different are identified, namely impervious land (ImpL), building facet (BudF), vegetated land (VegL), bare soil (BsoL) and water surface (WatS).

| Models | Surface energy balance | | | | | | | | | | | | | | | AH |
|---|---|---|---|---|---|---|---|---|---|---|---|---|---|---|---|---|
| | Shortwave and longwave radiation | | | | | Sensible heat flux | | | | | Latent heat flux | | | | | |
| | ImpL | BudF | VegL | BsoL | WatS | ImpL | BudF | VegL | BsoL | WatS | ImpL | BudF | VegL | BsoL | WatS | ImpL |
| SUEWS | Y | Y | Y | Y | Y | Y | Y | Y | Y | Y | Y | Y | Y | Y | Y | Y |
| VUCM | Y | Y | Y | Y | N | Y | Y | Y | Y | N | Y | N | Y | Y | Y | Y |
| ASLUM | Y | Y | Y | Y | N | Y | Y | Y | Y | N | Y | Y | Y | Y | Y | Y |
| TEB | Y | Y | Y | Y | N | Y | Y | Y | Y | N | Y | Y | Y | Y | Y | Y |
| UT&C | Y | Y | Y | Y | N | Y | Y | Y | Y | N | Y | Y | Y | Y | Y | Y |
| VCWG | Y | Y | Y | Y | N | Y | Y | Y | Y | N | Y | Y | Y | Y | Y | Y |
| BEP | Y | Y | Y | N | N | Y | Y | Y | N | N | N | N | N | N | N | Y |
| TARGET | Y | Y | Y | Y | Y | Y | Y | Y | Y | Y | N | N | N | Y | Y | Y |
| VTUF-3D | Y | Y | Y | Y | N | Y | Y | Y | Y | N | Y | Y | Y | Y | N | Y |
| Solene-Microclimat model | Y | Y | Y | Y | Y | Y | Y | Y | Y | Y | Y | Y | Y | Y | Y | Y |
| SWMM | N | N | N | N | N | N | N | N | N | N | N | N | N | N | N | N |
| Multi-Hydro model | N | N | N | N | N | N | N | N | N | N | N | N | N | N | N | N |
| URBS | N | N | N | N | N | N | N | N | N | N | N | N | N | N | N | N |
| WEP | Y | Y | Y | Y | Y | Y | Y | Y | Y | Y | Y | Y | Y | Y | Y | Y |

model for addressing the energy balance of the water surface within the urban canyon. This model draws inspiration from the pan evaporation model (Molina Martínez et al., 2006) and closely resembles the lake model proposed by Jacobs et al. (1998).

The Solene-Microclimat model utilises a 3D representation of urban geometry. Within this model, simulated scenes are created by meshing the 3D-developed surface with rectangular or triangular elements (Musy et al., 2015). The size of the simulated scene determines the area covered by these elements. The thermal-radiation model addresses the energy balance for each facet. The Solene-Microclimat model employs a "radiosity" method to compute multiple reflections. This method involves two critical aspects: 1) geometric form factors between all facets of the built surface and the sky vault are calculated using the

contour integral technique; 2) radiative properties (reflection, transmission, and absorption) are considered for all surfaces within the scene. The thermal-radiation model is coupled with a CFD (Computational Fluid Dynamics) model, allowing for the exchange of sensible heat fluxes that impact the energy balance, moisture mass transfer, near-surface air velocity, and air temperature. The Solene-Microclimat model also accounts for both trees and water surfaces (Musy et al., 2015, 2021). Trees are treated as surfaces in the radiative model (specifically, the tree crown envelope) and as volumes in the CFD model, considering

factors such as shading, solar radiation absorption, and aerodynamic resistance. The latent heat flux from vegetation is regulated by an evaporation rate, given the absence of a hydrological model. Additionally, the energy balance of water pond surfaces is based on a thermal model developed by Inard et al. (2004), which considers radiation absorption, transmission, and reflection at the water surface. Furthermore, the model includes the simulation of the water surface, as discussed by Musy et al. (2021).

    Regarding UHMs, energy balance simulations diverge significantly from ULSMs. In SWMM, energy balance processes are

notably absent. However, efforts have been made to integrate energy balance processes with hydrology in the water and energy transfer processes (WEP) model. Land use is initially divided into three groups in WEP: a water body group, a soil–vegetation group, and an impervious area group (Figure 7). The energy balance is solved for each type of surface. Short-wave radiation is based on observation or deduced from sunshine duration, and long-wave radiation is calculated according to temperatures. The interaction of radiation between soil and vegetation is considered by using the fraction of transmitted short-wave radiation of

vegetation, whereas the interaction between the urban cover and the urban canopy is considered by using the sky view factor of urban cover. Latent and sensible fluxes are computed using the aerodynamic method, and surface temperature is solved using the force–restore method (Jia et al., 2001).

    Conclusions can be drawn from the comparison and analysis of the surface energy balance among the selected models: Despite differing resolutions, both the bulk model SUEWS and the building-specific Solene-microclimate model account for

the surface energy balance across diverse landscapes. The majority of urban canopy models (UCMs) lack the capability to simulate water surfaces within urban canyons. Urban Land Surface Models (ULSMs) emphasise radiation and sensible heat processes, such as the obstruction and reflection of radiation by trees, while neglecting latent heat flux. Among the chosen hydraulic and hydraulic-hydrological models, only the WEP hydrological model is equipped to address the surface energy balance.

**Table 5.** Selected model capabilities concerning the near-surface hydro-meteorological processes. 'P' stands for 'profile', which indicates the model resolves vertical profiles; 'L' stands for 'Lumped', representing canyon average conditions; 'N' indicates models that do not resolve the urban canyon.

| Models | Urban canopy hydro-meteorology condition | | |
| --- | --- | --- | --- |
| | Near-surface air temperature | Near-surface humidity | Near-surface wind |
| SUEWS | P | P | P |
| VUCM | L | L | P |
| ASLUM | L | L | P |
| TEB | L | L | P |
| UT&C | L | L | P |
| VCWG | P | P | P |
| BEP | P | P | P |
| TARGET | L | L | P |
| VTUF-3D | L | L | P |
| Solene-Microclimat model | P | P | P |
| SWMM | N | N | N |
| Multi-Hydro model | N | N | N |
| URBS | N | N | N |
| WEP | L | L | P |

## 5.2 Urban canopy meteorology condition

All the models in the current study include atmospheric conditions but with different levels of complexity; the summary is displayed in Table 5. In the context of most UHMs, meteorological variables serve as forcings. They are shown as the 'N' in Table 5, representing that the model does not resolve the near-surface urban meteorology condition. Take the Storm Water Management Model (SWMM) as an example: it relies on precipitation, air temperature, and wind speed data. Precipitation data can come from multiple sources, such as nearby rain gauges or radar observations. When simulating snow melt or using the Hargreaves method to compute potential evaporation, SWMM also requires air temperature data. Wind speed, on the other hand, primarily affects the snowmelting process. However, due to the absence of an energy balance process, many hydrological models struggle to represent meteorological conditions within urban canyons accurately. The meteorological forcings for the hydrological models represent only the average conditions of the entire area considered.

In fact, the regional scale meteorological conditions are also part of the boundary conditions of the ULSMs. For the ULSMs within regional climate models, like WRF-SUEWS, WRF-SLUCM, and WRF-BEP, regional scale meteorological conditions are two-way coupled with ULSMs. Stand-alone ULSMs, in general, need the first-level air temperature, humidity, wind speed,

and precipitation as the input, together with radiation. ULSMs resolve the urban canopy atmospheric conditions with different complexities: canyon average (Table 5: 'L') and canyon profile (Table 5: 'P').

Six of ten included ULSMs solve the average canyon temperature and humidity. It is worth mentioning that the hydrological model WEP, which includes the energy balance, can also solve the canyon average meteorological conditions. The solution is based on the energy balance and is directly linked with turbulent heat fluxes among surfaces, canyons, and the upper atmosphere (Jia et al., 2001). Masson (2000) and Wang et al. (2013) documented in detail the scheme used to calculate the canyon mean temperature, humidity, and wind for single-layer UCMs. In the VTUF-3D model, although resolving radiative exchanges in 515  3D, the convection has been simplified into a 1D equation instead of more computationally intensive methods such as large eddy simulations (LES) or Reynolds-averaged Navier-Stokes (RANS) (Krayenhoff and Voogt, 2007). The convection scheme is also based on what is mentioned in Masson (2000), which is one-dimensional. For the regional climate models that are two-way coupled with USLMs, 2-meter air temperature, humidity, and 10-meter wind can be obtained for a single grid on average (Chen et al., 2011). Some recent studies applied a revised scheme based on sensible heat flux from urban canopy from 520  the ULSM in the regional climate model to specifically focus on 2-meter air temperature over urban areas (Chen et al., 2021; Theeuwes et al., 2014). In all selected ULSMs, the wind speed profile is assumed to be logarithmic above the urban canopy, exponential within the urban canyon, and logarithmic again close to the ground surface (Masson, 2000).

    The remaining four ULSMs are SUEWS, BEP, VCWG and the Solene-Microclimat model, which can resolve the vertical profile of air temperature, humidity, and wind within urban canyons. It is well known that the Monin-Obukhov Similarity Theory (MOST) provides the dimensionless mean flow and mean temperature in the surface layer under non-neutral conditions, which can be applied primarily to flat and horizontally homogeneous terrain (Högström, 1996). However, MOST fails to be applied in a layer extending from the ground to several canopy heights, the roughness sublayer (RSL), which can be two to five times the mean canopy height (Fazu and Schwerdtfeger, 1989; Moriwaki and Kanda, 2006). Several corrections for MOST have been proposed to resolve the conditions within the roughness sublayer, including those of Harman and Finnigan 530  (2007) and De Ridder (2010). Theeuwes et al. (2019) evaluated these two correction approaches that parametrise wind speed and temperature vertical profiles within the urban roughness sublayer. In the 2020a version of SUEWS, a diagnostic RSL scheme based on Harman and Finnigan (2007, 2008) for calculating the wind, temperature, and humidity profiles has been implemented. Researchers have attempted to quantify the interaction between urban elements and the atmosphere in relation to dynamic and thermal effects. This approach is grounded in the multi-layer model (referred to as BEP) (Martilli et al., 2002).

In contrast to MOST, which relies on gradient relationships, multilayer models consider the aerodynamic and thermal impacts of urban elements as either sources or sinks for potential temperature, momentum, specific humidity, and turbulent kinetic energy equations. In both BEP and VCWG, a one-dimensional vertical diffusion model that is based on the diffusion coefficient is applied to solve the vertical momentum, heat, and moisture fluxes (Santiago and Martilli, 2010; Martilli et al., 2002; Moradi et al., 2021). The major difference between BEP (multi-layer urban canopy model) and VCWG is that BEP solves the 540  energy balance for each layer of the urban canopy, whereas the radiation model in VCWG adopts the scheme of ASLUM. Until recently, the vertical diffusion model was in the developmental phase, with ongoing sensitivity tests aimed at enhancing the parameterised scheme. These tests were based on computational fluid dynamics (CFD) studies conducted by Nazarian et al.

(2020) and Simón-Moral et al. (2017). Unlike the other three models, the Solene-Microclimat model is a CFD model. Thus, 3D conditions can be obtained over the 3D simulation domain.

It can be concluded that, similar to their capacity for resolving the surface energy balance, Urban Land Surface Models (ULSMs) prioritise assessing near-surface urban hydro-meteorological conditions compared to Urban Hydrological Models (UHMs). All the selected ULSMs can resolve the vertical wind profile based on the similarity theory and can solve the 2D average (lumped) air temperature and humidity. For the bulk model, SUEWS, vertical profiles of atmospheric variables are included, but only for the grid average; the multi-layer urban canopy models (BEP and VCWG) and building resolving models

can resolve the vertical profile within the urban roughness sub-layer (urban canyon).

## 5.3  Surface water balance

Table 6 presents the capability of the included models to simulate the urban surface water balance, in which surface runoff, depression storage (water stored on the land surface depending on surface characteristics), infiltration, evaporation, snow-related processes, and irrigation are included. The complexity of the surface water balance-related processes is different per

model. Overall, the VUCM, BEP, TARGET, VTUF-3D, and Solene-Microclimat models do not have a good capacity for simulating the water balance over heterogeneous urban landscapes. Runoff only occurs when interception water of each surface type exceeds the maximum reservoir capacity. Interception storage is determined by the difference between the incoming source, which includes precipitation, and the sink, which includes evaporation and infiltration. Different models include various source and sink terms. For example, irrigation is included in SUEWS, ASLUM, TEB, VCWG and UT&C, but leakage from

impervious roofs is considered an extra sink term in UT&C and VCWG, and the snow melting process is also only included in SUEWS and TEB (Järvi et al., 2014; Colas et al., 2024). The sub-sections below compare the other processes.

### 5.3.1  Runoff

Many of the ULSMs assume that runoff travels directly to the sewer system and does not affect the water balance in the urban canyon; these models include VUCM, ASLUM, TEB, UT&C, and VCWG (Lee and Park, 2008; Wang et al., 2013;

Stavropulos-Laffaille et al., 2018; Meili et al., 2020; Moradi et al., 2022). Based on this assumption, there is no limit for surface runoff to enter the sewer system. It is worth mentioning that in UT&C and VCWG, a fraction of the surface runoff is maintained in the system for depression storage (ponding) and exchanges between the various surfaces in the urban environment before the water reaches the sewer system, which is a source term for the next simulation time step (Meili et al., 2020; Moradi et al., 2022). For example, the models can account for runoff from impervious areas that is redirected to infiltrate over bare land

areas. The rest of the surface runoff follows the assumption that it disappears in the water balance. In contrast, a limit to pipe inflow is considered for runoff in SUEWS. If the pipe capacity is exceeded, flooding above ground occurs as the water is added to above-ground runoff. The total runoff is the sum of the runoff to the pipe system and the above-ground runoff in SUEWS (Järvi et al., 2011).

The reservoir concept solves runoff in UHMs. In WEP, surface water runoff is estimated as precipitation minus evaporation

(Jia et al., 2001). In impervious areas, surface runoff can be obtained from a water balance analysis including depression

**Table 6.** Selected model capabilities regarding simulation of the surface water balance process over different surfaces. 'Y' represents that the process is included; 'Y-av' represents indifference among the land use types, and 'N' represents the exclusion of the process. Different surfaces are considered, namely impervious land (ImpL), building facet (BudF), vegetated land (VegL), bare soil (BsoL) and water surface (WatS).

| Models | Runoff | | | | Depression storage | | Infiltration | | Evaporation | | | | | Snow melting | Snow removing | Irrigation |
|---|---|---|---|---|---|---|---|---|---|---|---|---|---|---|---|---|
| | ImpL | BudF | BsoL | VegL | ImpL | BsoL | ImpL | BsoL | ImpL | BudF | VegL | BsoL | WatS | | | |
| SUEWS | Y | N | Y | Y | Y | Y | N | Y | Y-av | Y-av | Y-av | Y-av | Y-av | Y | Y | Y |
| VUCM | Y | N | N | N | N | Y | N | Y | Y | N | Y | Y | N | N | N | N |
| ASLUM | Y | Y | Y | Y | Y | Y | N | Y | Y | Y | Y | Y | N | N | N | Y |
| TEB | Y | Y | Y | Y | Y | Y | Y | Y | Y | Y | Y | Y | N | Y | Y | Y |
| UT&C | Y | Y | Y | Y | Y | Y | N | Y | Y | Y | Y | Y | N | N | N | Y |
| VCWG | Y | Y | Y | Y | Y | Y | N | Y | Y | Y | Y | Y | N | N | N | Y |
| BEP | N | N | N | N | N | N | N | N | N | N | N | N | N | N | N | N |
| TARGET | N | N | N | N | N | N | N | N | N | N | N | N | Y | N | N | N |
| VTUF-3D | N | N | N | Y | N | N | N | N | N | Y | Y | N | N | N | N | N |
| Solene-Microclimat model | N | N | Y | Y | N | Y | N | Y | N | Y | Y | Y | Y | N | N | N |
| SWMM | Y | Y | Y | Y | Y | Y | N | Y | Y-av | Y-av | Y-av | Y-av | N | Y | N | N |
| Multi-Hydro model | Y | N | Y | Y | Y | Y | N | Y | N | N | N | N | N | Y | N | N |
| URBS | Y | Y | Y | Y | Y | Y | Y | Y | Y-av | Y-av | Y-av | Y-av | N | N | N | N |
| WEP | Y | Y | Y | Y | Y | Y | N | Y | Y | Y | Y | Y | Y | N | N | Y |

storage, precipitation, and evaporation on land surfaces. It is assumed that there is no infiltration in impervious areas. The infiltration is included for bare land in WEP. WEP is a grid-based model, which means it solves the energy and water balance for each grid cell. The subgrid heterogeneity of land use is considered by using the mosaic method. The main structure of WEP is similar to most of the ULSMs, but WEP simulates the 1D river flow horizontally. River flow routing is conducted for every tributary and a main channel by using the kinematic wave method. Overland flow is simplified as lateral inflow to rivers. In URBS, the surface is represented by an interception reservoir (Rutter et al., 1971), whose capacity depends on the land use type: natural soil, streets, or roofs. The non-intercepted rainwater is separated into three components: evaporation, infiltration into the vadose zone, and surface runoff. Surface runoff occurs only when the surface storage exceeds its capacity. The URBS model solves the water balance at the urban hydrological element (UHE) level (Berthier et al., 2004; Rodriguez et al., 2003). The representation of the UHE is also 2D, which is similar to the representation in the SUEWS. However, URBS considers the spatial discretisation based on UHE connected to the drainage network. Thus, the horizontal spatial distribution can be solved. In SWMM, the reservoir concept for generating runoff is more complex, called the nonlinear reservoir model (Rossman and Huber, 2016). The most obvious difference in the nonlinear reservoir, as indicated by its name, is that the runoff volume is not linearly linked to the water depth above the depression storage depth. Further, it is also related to the surface roughness coefficient, the apparent or average slope of the sub-catchment, and the area across the sub-catchment's width through which the runoff flows. Each sub-catchment separates the land into categories: pervious land area and impervious land with and without depression storage. SWMM supports the overland flow within one sub-catchment among the pervious and impervious areas, as well as overland flow among sub-catchments. The overland flow can be solved by the Multi-Hydro model by equations ensuring the conservation of mass (continuity) and momentum. This flow depends on the surface properties as well as the elevation and is computed using the diffusive wave approximation of Saint-Venant equations (Ichiba et al., 2018).

Surface runoff is one of the most essential items in the water balance. Based on the comparison, it can be concluded that most of the selected models that can solve the water balance would address the runoff amount based on the reservoir concept. However, in the ULSMs, most of the runoff cannot participate in the water balance at the next time step, which is highly related to whether the model includes the sewer system capacity and run-on portion. Overland flow cannot be solved by all selected ULSMs but can be solved by some of the UHMs (SWMM and Multi-Hydro model).

### 5.3.2 Depression storage and infiltration

Depression storage and infiltration describe the process of the water held by the land surface and transferred downward through the surface. In ULSMs, depression storage and infiltration over impervious and bare soil land are included. Among the impervious land, the infiltration part is defined as a constant value that must be calibrated or ignored. TEB sets the infiltration rate over impervious roads, referring to Ramier et al. (2011). However, the infiltration over impervious land is not accounted for in VUCM, ASLUM, UT&C, and VCWG. Several methods in these models simulate infiltration and depression storage on bare land. The 1D Richards equation is applied in some of the selected models, including ASLUM, UT&C, and VCWG (Wang et al., 2013; Meili et al., 2020; Moradi et al., 2022). The Richards equation is a differential equation that relates soil moisture content to hydraulic conductivity and capillary pressure. It accounts for both vertical flow driven by the hydraulic gradient

and capillary rise. In TEB and SUEWS, the Green and Ampt model is adopted (Järvi et al., 2011; Stavropulos-Laffaille et al., 2018). The Green-Ampt model is a simplified physical model based on the Richards equation. It assumes that infiltration occurs instantaneously at the soil surface (Kale and Sahoo, 2011). One more option in SUEWS for infiltration simulation is the Horton method. The Horton method is an empirical model. It assumes that infiltration decreases exponentially over time until it reaches a constant value, typically the saturated hydraulic conductivity (Horton, 1941).

For the UHMs, the Richards equation and Green-Ampt model are also adopted by the distributed WEP, URBS, and Multi-Hydro models. In SWMM, there are four options for infiltration, including the Green-Ampt model, Horton's method and its modified version, and the Curve Number Method (Rossman and Huber, 2016). One major difference between the Curve Number Method and the Green-Ampt model is that the curve number model assumes an initial abstraction before the surface runoff, while the Green-Ampt model assumes surface runoff only when the precipitation rate is greater than the infiltration rate. There is no universal conclusion on the best method. Previous studies have conducted a comparison analysis among different infiltration methods for various cases (Parnas et al., 2021; Mallari et al., 2015).

Generally, if the model includes the water balance, it considers the depression storage and infiltration process. Based on the above summary, depression storage over impervious land areas can be solved relatively easily. The infiltration process over impervious land is not taken into consideration because the saturated hydraulic conductivity of these surfaces (e.g., roofs, paved streets) is fundamentally zero.

### 5.3.3 Evaporation and transpiration

The simulation of evaporation and transpiration is closely linked to latent heat flux, which connects the energy and water balances. These processes can be represented by latent heat flux (watts per square meter) in the energy balance and evapotranspiration (ET) in the water balance (water depth per unit time). The Penman and Penman-Monteith equations are used to quantify moisture and latent heat flux, applicable in Urban Land Surface Models (ULSMs) and Urban Hydrological Models (UHMs). Despite using the same methodology, different models vary in how they account for evaporation and transpiration in urban environments. Table 7 details evaporation and transpiration in the water balance, while Table 4 shows models including latent heat flux.

In ULSMs, the energy balance can be independent of the water balance. For example, BEP excludes the water balance and latent heat flux. Other ULSMs simplify evaporation and transpiration over impervious land compared to vegetated or bare land. The method for simulating latent heat flux (LH) from impervious land is documented by Masson (2000) and Wang et al. (2013). When water depth on impervious land is known, LH and ET are calculated based on specific humidity, saturated specific humidity at surface temperature, and aerodynamic resistance, with water depth updated via the water balance equation. For bare land, a reduction factor based on soil water content is used and updated at each simulation step.

Evaporation from vegetation is complex due to differences between short and tall vegetation. In ASLUM and TEB, stomatal resistance is considered for vegetated land, using a parameterization scheme based on meteorological variables like solar radiation, soil-water availability, vapor pressure deficit, and air temperature. This method suits short ground vegetation but does not account for the sun's direction. VTUF-3D and UT&C simulate evaporation and transpiration in detail, with transpiration

modeled through stomatal conductance dependent on plant photosynthesis. UT&C also considers evaporation from intercepted water on vegetation canopy and transpiration from sunlit and shaded ground vegetation and tree canopy (Meili et al., 2020; Krayenhoff and Voogt, 2007). VCWG adopts much of UT&C's hydrological module (Moradi et al., 2022). The TARGET model only considers latent heat flux from water surfaces, not other urban surfaces (Broadbent et al., 2019). VTUF-3D combines TUF-3D for impervious land and MAESPA for vegetated land, considering the water balance and latent heat flux only for vegetated land (Krayenhoff and Voogt, 2007).

There are several groups of methods to calculate evaporation. The first group includes SUEWS (Ward et al., 2016) and WEP (Järvi et al., 2011), which are based on the Penman-Monteith equation. These models initially determine the net radiation, which is the sum of the heat fluxes. They then calculate the storage heat flux and anthropogenic heat flux, followed by the latent heat flux using the Penman-Monteith equation. The energy balance closure is achieved by setting the residual as the sensible heat flux. The second group comprises VUCM, ASLUM, TEB, UT&C, and VCWG (Lee, 2011; Meili et al., 2020; Masson, 2000; Wang et al., 2013), which use the resistance method. This method directly calculates the moisture heat flux based on the saturation value of specific humidity at a given surface temperature, solving the energy balance and surface temperatures directly. The third group, represented by URB (Berthier et al., 2004), assumes that the evaporation flux is proportional to the potential evapotranspiration and water storage, with the potential evapotranspiration being an input forcing for the model. Lastly, SWMM (Rossman, 2017) offers three simplified approaches for modeling: assigning a single constant value, utilizing input data, or deriving daily values from temperature data provided in an external climate file. Additionally, SWMM allows users to specify a set of "drying times" based on Horton's method, which are independent of climatological variables. This method estimates the transition from a "fully saturated" to a "fully dry" state, effectively incorporating both percolation and evaporative losses into a single term. This suggests that UHMs prioritize accurate runoff quantification, treating evaporation primarily as a generalized loss component.

It is clear that, although at different levels of complexity, many ULSMs include the linkage between the thermal environment and water availability through the evaporation and transpiration process with a two-way coupling. However, in most UHMs, the temperature used for calculating the evaporation and transpiration process is only input data, as mentioned in Section 5.2. The Multi-Hydro model ignores the water loss from the evaporation and transpiration process in the water balance. In SWMM, several options can yield the ET rate:

(1) Set as a single constant value that does not have temporal variation. The value can also be set based on the simulation month, and the user can also set daily temporal information.

(2) Read from an external climate file.

(3) Compute the daily values from the daily temperatures in an external climate file.

The calculation in SWMM adopts the Hargreaves method (Hargreaves and Samani, 1985), which is based on daily max-min temperatures contained in a climate file and the study area's latitude. The latitude sets the radiation water equivalent of incoming extraterrestrial radiation. This method is suitable for the case without detailed information on the latent heat flux being included.

WEP and URBS share a similar approach to the ULSMs based on the Penman-Monteith equation. In WEP, evaporation over water surfaces is accounted for considering the psychometric constant. ET from vegetation in WEP is a combination
of interception evaporation, transpiration from low vegetation, and from high vegetation (Jia et al., 2001). Since the energy balance is calculated by WEP, the latent heat flux (LH) is calculated based on net radiation and ground heat flux and linked with the ET. WEP is also the only UHM that is included that considers differences in evaporation and transpiration processes for different land uses. However, for URBS, meteorological conditions and potential evaporation are inputs for the model. The evaporation flux is assumed to be proportional to the Penman–Monteith potential evapotranspiration and water storage
(Rodriguez et al., 2008).

Evaporation, as one of the sink terms in the water balance, is considered by most of the ULSMs that have the ability to solve the water balance but can be ignored by some UHMs due to the small amount of water lost through the process. Most of the ULSMs focus less on evaporation over the water surface. Furthermore, ULSMs solve the evaporation over the different landscapes separately within a simulation unit, but UHMs usually solve the process over the simulation unit as a whole.

## 5.4  Urban subsurface

ULSMs and UHMs resolve several hydro-meteorological processes beneath the ground surface. Water stored in the soil layers below the surface can be transported upward through transpiration. Five of the selected models, SUEWS, VUCM, BEP, TARGET, and VTUF-3D, do not include this process. The two sub-sections below discuss the moisture conditions in soil layers in the subsurface and the processes related to the (man-made) pipe system.

### 5.4.1  Moisture transfer in and between soil layers

For most of the included ULSMs, a multi-layer soil hydrology model is included. Exceptions are SUEWS and TARGET. SUEWS includes a bucket soil model, while TARGET does not include a hydrology module at all (also mentioned in Section 5.3). In VUCM, the moisture flux between the soil layers is calculated based on the hydraulic conductivity of the soil, soil moisture content, and the depth of the soil layer (Clapp and Hornberger, 1978). The moisture flux at the bottom is set to zero. It
is worth mentioning that surface runoff is not parameterised over the vegetated land in VUCM. Instead, when the water content on the surface of canyon vegetation exceeds the maximum amount that vegetation can hold, the excess amount is first brought to thermal equilibrium with the vegetation temperature by heat transfer and then shed from the vegetation to the soil surface. In ASLUM, infiltration and runoff are both considered to occur over bare land and vegetated land. Soil moisture in the first layer is solved by the surface water balance, and the soil layers below are solved by the water exchange among the nearest
two layers. Soil-water diffusivity and hydraulic conductivity for computing the infiltration into the soil layer are determined using a method based on Clapp and Hornberger (1978). The moisture flux at the bottom is assumed to be zero, meaning no deep drainage. In both VUCM and ASLUM, no soil layers beneath impervious areas exist, in contrast to TEB and UT&C. The parameterisations of the vadose soil moisture dynamics are more complete in TEB and UT&C compared to the other models, with both vertical and lateral moisture fluxes considered. Vertical moisture fluxes are simulated by the 1D-Richards equation,
which is similar to those in VUCM and ALUCM. The lateral moisture flux in the upper soil layers is computed solely between

**Table 7.** Selected model capabilities regarding simulation of the urban subsurface conditions. The complexity of resolving soil moisture can be single (SL) or multi-layer (ML), and the water transfer among the soil layer(s) can be vertical (Vf) or/and lateral within/between the grid(s) (Lfin/Lfbt). The presentation of the pipe system can be only the inflow with or without limitation (Inflow - limited/Inflow - infinite), or it can be with a detailed hydraulic system (Hydraulic). ImpL stands for impervious land and BsoL stands for bare soil land.

| Models | Urban subsurface | | | |
| --- | --- | --- | --- | --- |
| | Soil layers moisture | | Transpiration | Pipe system |
| | ImpL | BsoL | | |
| SUEWS | SL - Lfin | | N | Inflow - limited |
| VUCM | N | ML - Vf | N | Inflow - infinite |
| ASLUM | N | ML - Vf | Y | Inflow - infinite |
| TEB | ML - Vf - Lfin | | Y | Inflow - infinite |
| UT&C | ML - Vf - Lfin | | Y | Inflow - infinite |
| VCWG | ML - Vf - Lfin | | Y | Inflow - infinite |
| BEP | N | N | N | N |
| TARGET | N | N | N | N |
| VTUF-3D | N | N | N | N |
| Solene-Microclimat model | N | ML - Vf | Y | Hydraulic |
| SWMM | ML - Vf - Lfbt | | Y | Hydraulic |
| Multi-Hydro model | ML - Vf | | Y | Hydraulic |
| URBS | ML - Vf - Lfin | | Y | Hydraulic |
| WEP | ML - Vf - Lfbt | | Y | Inflow - infinite |

the garden and building compartments in TEB and is similar in UT&C in that the horizontal water transfer is absent at the top two layers of soil. It is worth mentioning that, although SUEWS uses a single-layer soil scheme, lateral flow is considered. The soil hydraulic conductivity and soil water potential in TEB and SUEWS are based on Clapp and Hornberger (1978) and van Genuchten (1980), respectively. In UT&C, the soil hydraulic properties can follow either the van Genuchten (1980) or the

Saxton and Rawls (2006) parameterisation. Moreover, the transpiration sinks of high and low vegetation are included while calculating the soil layer moisture in ASLUM, UT&C, and TEB, depending on the root biomass fraction in each soil layer. In TEB, the infiltration to the pipe system from saturated soil moisture is further represented, which is adapted from URBS, one of the selected hydraulic-hydrological models (Rodriguez et al., 2008).

Because SWMM was originally developed to simulate combined sewer overflows in urban catchments, the fate of infiltrated

water was considered insignificant. The subsurface soil layers are represented by a two-layer scheme: an unsaturated upper zone and a saturated lower zone, which is called a two-zone groundwater model. In SWMM, three options can be selected: Green–Ampt model, Horton's method and Curve number method. The scheme in URBS is similar to SWMM in that it has two zones. The vertical transfer between the unsaturated upper zone and the saturated lower zone is estimated to be similar to

those in the ULSMs, but instead of the 1D-Richards equation, the Green–Ampt model is applied. The infiltration to artificial networks from saturated soil moisture that is included in TEB is also included in URBS. In SWMM, lateral outflows in and out of the conveyance system are further considered in the saturated zone. The top three layers in WEP are called the subsurface in a narrow sense. The water balance among each subsurface is calculated differently. The continuity of water movement is considered when the application of the generalised Green–Ampt model is switched to that of the Richards model and vice versa (Jia et al., 2001). Subsurface runoff (lateral flow) is calculated according to land slope and soil hydraulic conductivity. The capillary rise can be a source term of the upper unsaturated soil layers in WEP and is parameterised by means of Boussinesq equations (Jia et al., 2001). Soil conductivity and saturated soil conductivity in urban hydrological models are consistent with those represented in most ULSMs. These parameters can be calibrated in the field using infiltrometers and laboratory measurements, such as the constant head and falling head methods (Gupta et al., 2021). Additionally, they can be estimated using empirical models based on soil type, moisture content, and temperature, such as Pedotransfer Functions (Van Looy et al., 2017). The accuracy of these estimates can be further enhanced through machine learning techniques, including artificial neural networks and random forests (Jian et al., 2021).

By comparing all the selected ULSMs and UHMs, we found that multi-layer soil moisture and vertical water transfer are commonly solved for bare soil land in the ULSMs capable of simulating the water balance and in UHMs. The lateral water transfer between the simulation units exists only in UHMs. The development of integrating hydrological processes in the subsurface in ULSMs would follow the methods used in UHMs.

### 5.4.2  Pipe system

Pipe inflow, also called sewer discharge, is usually equal to the total runoff and 'disappears' from the further water balance in most ULSMs, including VUCM, ASLUM, TEB, UT&C, and VCWG (Lee and Park, 2008; Wang et al., 2013; Stavropulos-Laffaille et al., 2018; Meili et al., 2020; Moradi et al., 2022). In most cases, there is no limit to this discharge. Only in SUEWS, groundwater runoff can be distinguished from pipe inflow, and there is a limit to the amount of water that can enter the pipe system (Järvi et al., 2011). This is consistent with the conclusions mentioned in Section 5.3.1.

None of the included ULSMs follow the pipe system scheme. This would be called a hydraulic model and is the core of SWMM. The SWMM hydraulics model includes seven parts: junction nodes, outfall, dividers, storage units, conduits, pumps, and regulators (Rossman and Simon, 2022). Some hydraulics processes that are included in SWMM, such as external inflow of surface runoff, groundwater interflow, rainfall-dependent infiltration/inflow, and non-uniform flow routing through any configuration of open channels, pipes, and storage units, are closely coupled with hydrological processes like surface runoff, evaporation, and infiltration. The governing equation for the water movement through a conveyance network of pipes and channels is the St. Venant equation for unsteady free surface flow. In the distributed Multi-Hydro model, the hydraulic model is adopted from SWMM and is coupled two-way with the surface module and soil module. In hydraulic-hydrological models, the hydraulic part is usually not as complete as in a hydraulic model such as SWMM. The drainage module in the Multi-Hydro model is based on the 1D SWMM. In URBS, the hydraulic routing configuration has been modified to contain two modelling stages: routing of surface runoff on streets from UHEs up to the sewer inlets, as represented by the travel

time routing procedure used in URBS, and hydraulic routing inside sewer networks, as represented by using the classical Muskingum–Cunge scheme, which offers an approximate solution to the diffusive wave equation (Rodriguez et al., 2003). WEP is the only UHM that considers the pipe inflow and assumes the inflow amount is infinite without considering pipe capacity, which is similar to ULSMs.

Based on the above summary, it is obvious that the pipe system in ULSMs is not well represented in the water balance calculation. The pipe system includes various levels of detail depending on the model focus of the UHMs.

## 6 Challenges and future developments

### 6.1 Urban thermal environment adaptation

The thermal environment depends not only on temperature, but also on humidity. Thus, accurately estimating these two terms is essential for the investigation. Two directions of efforts have been made in the last decades on ULSMs.

First, ULSMs are widely implemented in regional climate models, as mentioned in Section 3 and Figure 3. Coupling the atmospheric models with land surface models, including ULSMs, supports studies of land-atmosphere interactions, specifically the impact of urban land on this interaction. Recent studies have further employed physically realistic, coupled atmosphere-land surface-subsurface models (Wagner et al., 2016; Fersch et al., 2020; Kim et al., 2021). These coupling simulations aim to evaluate the intricate interplay between terrestrial and atmospheric processes, shedding light on their influence on hydro-meteorological phenomena. By coupling WRF-SLUCM and a land surface-subsurface model (ParFlow), Talebpour et al. (2021) emphasised the critical need to account for terrestrial hydrology, particularly in urban areas where diverse development patterns introduce additional complexities to coupled atmosphere-groundwater interactions. The coupling between the WRF-SLUCM and ParFlow can overcome some of the limitations of representing the saturated soil layer and groundwater in ULSMs found in the current study.

Second, the ULSMs are further developed, accounting for complex physical processes, like hydrological processes, vegetation representation, to support studies on the interface of meteorology and hydrology. In general, the development of ULSMs initially focused on stand-alone models, which have subsequently been implemented in coupled versions with mesoscale climate models. Therefore, stand-alone versions are usually more advanced in physical complexity than models in the coupled version. This is the case for, e.g., ASLUM (Wang et al., 2013) and WRF-SLUCM (Yang et al., 2015), as well as the SUEWS and WRF-SUEWS (Sun et al., 2023; Sun and Grimmond, 2019). However, the hydrological module of BEP was first implemented in a coupled version by Yu et al. (2022) (WRF-BEP) instead of as a stand-alone version. In the version of WRF-BEP, the depression over the impervious land, evaporation over the urban green surface, and the pipe system parameterisation based on Manning's formula are included (Yu et al., 2022).

However, urban land surface models (ULSMs) still face challenges in capturing the interaction between the natural land surfaces within urban areas, including water and vegetation, and the built-up areas. These natural patches are crucial for simulating the urban thermal environment, affecting energy balance and humidity levels. Based on our study, stand-alone ULSMs have started incorporating natural land surfaces, including vegetated land and trees (Wang et al., 2021a; Krayenhoff

et al., 2020), but often exclude urban lakes and ponds. In current land-atmosphere coupled mesoscale simulations, land surface models (LSMs) simulate natural lands, while ULSMs focus exclusively on impervious urban areas. With the development of mesoscale simulation at hectometer (100-meter) scales, spatial resolution can be improved in coupled ULSMs, allowing small natural landscapes within urban areas to be captured (Lean et al., 2024). Urban surface waters receive less attention compared to vegetated areas. The representation of the permanent urban surface waters, i.e., urban lakes and urban ponds, is recommended to be implemented into ULSMs. We also recommend that the Local Climate Zone (LCZ) scheme consider urban water landscapes in its classifications. For the coupled ULSMs, adjustments are also needed for simulating urban lakes due to differences from natural lakes.

## 6.2 Urban flood forecasting

Accurate precipitation data and coupling the hydrological models with hydraulic models contribute to accurately forecasting urban flooding and waterlogging. Atmospheric models have been coupled one-way with urban hydraulic models. The coupling enables more precise monitoring and control of hydraulic systems, thereby enhancing the accuracy of flood predictions and responses. Traditionally, hydraulic models utilise single-grid precipitation data as input; however, understanding the urban impact on precipitation events reveals significant spatial variability within a catchment (Cristiano et al., 2017). Consequently, higher-resolution input data from the atmospheric model can improve hydraulic system performance. Furthermore, the coupling can more effectively manage waterlogging by providing real-time data on soil saturation and atmospheric conditions, leading to improved predictions and responses to waterlogging events. For example, recently Gu et al. (2022) coupled WRF with SWMM to provide precipitation input for the hydro-hydraulic simulation. Furthemore, SWMM solves the one-dimensional Saint-Venant equations, which limits its effectiveness for modeling urban flood propagation. To address this, integration with IBER—a two-dimensional hydraulic model—can be employed to develop a coupled 1D-2D model, where the 1D component represents the piped drainage network and the 2D component captures overland flow (Sañudo et al., 2025).

Urban land modifies both atmospheric and terrestrial hydrological processes. By leveraging the two-way coupling between land and atmospheric models, researchers can explore the effects of urbanisation on atmospheric and terrestrial hydrological processes, respectively. For instance, an emerging study area involves investigating urban-induced convective precipitation events (Wang et al., 2021b; Yang et al., 2024). Accurately observing and simulating high-resolution rainfall are challenging due to the spatial variability and randomness in rainfall events. There is another group of existing studies using hydrological models to investigate the impact of urbanisation on terrestrial-hydrological processes, such as runoff and infiltration (Oudin et al., 2018; Locatelli et al., 2017; Yoo et al., 2021). However, these investigations often overlook atmospheric interactions, concentrating solely on land surface and subsurface hydrological processes. This alignment with the findings in the current analysis underscores the complexity of terrestrial-hydrological processes, which exhibit strong coupling with atmospheric dynamics.

To enhance urban flooding forecasts, it is crucial to integrate atmospheric, hydrological, and hydraulic models effectively. Accurate precipitation forecasts require a well-represented land surface in a two-way coupled land-atmospheric simulation. At the same time, the hydrological model WRF-Hydro shows potential for coupling with a land-surface model and an atmospheric

model. This can provide an opportunity to study the urban impacts on terrestrial-hydrological processes and their subsequent effects on atmospheric-hydrological interactions simultaneously. In the end, the hydraulic models take both outputs from the atmospheric and hydrological models to better simulate the urban flooding and waterlogging.

## 6.3 Compound extreme event analysis

Extreme weather events such as heavy rainfall, floods, droughts, and heatwaves can occur either simultaneously or in succession, resulting in compound events. Permanent urban water bodies like lakes and canals, along with temporary water accumulations on impermeable surfaces from heavy rainfall, and drought conditions characterized by low soil moisture and water levels, significantly affect the urban thermal environment by altering the energy budget and humidity levels (Hao et al., 2023). Additionally, convection generated by urban heat islands can influence the spatial distribution of convective rainfall, further affecting land hydrological processes. Investigating the interconnections and mutual influences of these compound events is crucial. Therefore, assessing the risk of extreme weather, particularly compound extreme events, requires understanding land-atmosphere interactions and the coupling of thermal and hydro-hydraulic processes. For example, this understanding can enhance the simulation of urban micro-climates during and after extreme precipitation events, optimizing drainage and mitigating adverse effects on local ecosystems.

Currently, coupling and integration are mainly applied to land surface and atmospheric models. Based on our findings, one essential point that prevents the hydraulic and terrestrial-hydrological processes from being fully implemented with the thermal processes in the simulation is that overland flow, channel flow, and subsurface lateral flow of water have to include the interaction and connection between the simulation units, which is the grid cell in most of the ULSMs, notably Multi-Hydro and WEP, and the sub-catchment in SWMM, UHE in URBS. A notable example of this approach is WRF-Hydro, which is coupled with a land surface model that includes terrain routing modules, allowing it to solve 2d overland flow with links among the simulation grids and soil moisture (Gochis et al., 2018). WRF-hydro also includes channel and reservoir routing modules and can be two-way coupled with the atmospheric model. WRF-hydro can be coupled with SMWW to account for urban hydrology and drainage (Son et al., 2023). However, based on our knowledge, the development and application of the WRF-Hydro model, focusing on urban areas and the interface between the urban thermal and hydrological environment, are limited (Fersch et al., 2020).

Advancements in computing systems have led to the use of CFD models, traditionally for urban wind analysis, in simulating urban thermal environments. Despite limitations in 3D ULSMs for hydrological cycles, they show potential for urban hydro-thermal environments when integrated with distributed hydrological models (Robineau et al., 2022). However, both 3D ULSMs and URBS hydrological models require high-resolution input data, such as urban morphology, topography, and sewer systems. This need for detailed urban data limits the application of URBS compared to models like SWMM. As data collection and measurement methods for urban morphology and infrastructure improve, high-resolution simulations of urban microclimates and hydrological processes at the neighbourhood scale may become feasible, enabling comprehensive thermal and hydro-hydraulic interactions. At the neighbourhood and microscale levels, CFD models offer significant advantages, which are better suited for studying mitigation measures for climate-hydrological processes and hazards at the community planning scale. Therefore,

further development of CFD models is needed to embed physical processes or integrate them with high spatial-temporal resolution hydraulic-hydrological models. The extent of landscape heterogeneity and complexity of physical processes to be included should be considered in terms of data availability and computational resources.

### 6.4 Collaboration within and across disciplines

Our discussion on model comparison and future numerical model development highlighted the potential and challenges of inter- and intra-disciplinary cooperation. Collaboration between atmospheric science, hydrology, and hydraulics is essential for both regional-scale research and micro-scale applications. Detailed terrain data, urban building information, and other model inputs can advance research on urban hydrological and climate processes. This collaboration extends to geology, civil engineering, and architecture. The concept of urban geoscience emphasizes the need for communication across geoscience fields to address climate and natural disasters (Bricker et al., 2024). Urban geoscience, which applies geology and earth sciences to urban management, includes disciplines like engineering geology, hydrogeology, geological modelling, geochemistry, and environmental geology. It focuses on interactions between urban environmental systems and human activities, aligning with our findings (Bricker et al., 2024; Pescatore et al., 2024).

To promote collaboration, establishing common modeling protocols is imperative. Standardization in numerical simulation offers benefits like consistency, quality control, efficiency, enhanced collaboration, interoperability, predictability, and innovation. The NetCDF Climate and Forecast (CF) Metadata Conventions facilitate sharing and processing climate and forecast data in NetCDF files (Hassell et al., 2017; Eaton et al.). These conventions ensure interoperability between datasets from different sources. A common framework ensures consistent processes and outputs, reducing variability and enhancing quality, saving time and resources. Standardized protocols improve collaboration among researchers, enabling different models and systems to work together for comprehensive cross-model comparisons. Findable, Accessible, Interoperable and Reusable (FAIR) guiding principles for scientific data management and stewardship were put forward by (Wilkinson et al., 2016), emphasizing the existing challenges of limited public visibility, lack of interdisciplinary collaboration, data scarcity, and geographical variability (van der Werf et al., 2025). The Urban-PLUMBER initiative benchmarks and evaluates land surface models used in urban hydroclimate simulations, outlining key components like data formats, experiments, and expected outputs (Lipson et al., 2024). These initiatives should be expanded and formalized into widely accepted community practices for urban hydro-meteorological simulations.

Building on the importance of interdisciplinary collaboration highlighted in our discussion, artificial intelligence (AI) and machine learning offer transformative potential for urban climate and environment modeling (Ravuri et al., 2021). These technologies can enhance predictive accuracy, identify complex patterns, and optimize model parameters, thereby complementing the collaborative efforts in atmospheric science, hydrology, and hydraulics. It has been shown that machine learning models and statistical models can work together with the physics-based models while being applied to different urban adaptation strategies under climate change (Li et al., 2022; Aliabadi et al., 2023). To study the urban heat, machine learning is applied for downscaling by generating high-resolution data from lower-resolution results from physics-based simulations, reducing temperature errors, and lowering computational costs (Wu et al., 2021b). Forecasting can benefit from AI and machine learning by work-

ing with numerical simulations and measurement data to enhance data accuracy and model performance for climate disasters (Luo et al., 2022). He et al. (2023) introduce a hybrid data assimilation and machine learning framework integrated into the WRF, which optimizes surface soil and vegetation conditions to improve regional climate simulations. To fully leverage these advancements, urban climatologists and hydrologists should also engage in discussions with scientists who specialize in AI applications. These interdisciplinary dialogues are crucial for integrating AI-driven insights into urban geoscience, aligning with our findings and recommendations on the necessity of interdisciplinary cooperation to address climate and natural disasters effectively.

To overcome barriers in collaborative model development, establishing a practical technical framework is essential. Open-source platforms like GitHub enhance transparency, modular development, and community engagement by allowing real-time collaboration. The Weather Research and Forecasting (WRF) model, widely adopted and flexible, can serve as a common base for hosting and testing integrated urban hydroclimate components. Most bulk and 2D ULSMs can be two-way coupled with WRF, though 3D ULSMs are less coupled. Emerging multi-physics urban large-eddy simulation models, such as PALM-4U (Gehrke et al., 2021; Resler et al., 2017), and uDALES (Owens et al., 2024) and City-LES(Kusaka et al., 2024), benefit from increased computer resources and show potential to work with WRF outputs, e.g., WRF-PALM (Lin et al., 2021). WRF links atmospheric, land surface, hydrological, and hydraulic models and can work with machine learning models (He et al., 2023). WRF's robust capabilities make it ideal for integrating various urban climate models, facilitating comprehensive simulations. Utilizing such frameworks and platforms can drive innovation and improve urban hydro-meteorological research quality at multiple scales.

## 7    Conclusions

In response to the growing significance of research on compound natural disasters, implementation of nature-based solutions in urban areas and their impact, we have reviewed key models for simulating urban meteorological and hydrological environments. These models are either widely used, rapidly developed, or highly specialised. By disassembling and comparing the structure of various models, the complexity of the physical processes considered, and the resolving methods, we have found that simulation tools for urban thermal and hydrological environments have evolved in parallel over time. Although starting from different objectives, the interface of urban hydro-meteorology is receiving increased attention due to the potential implications of climate change and the imperative of climate adaptation. Recent trends in interdisciplinary research highlight the growing importance and potential of collaborative efforts in atmospheric science, hydrology, and hydraulics for advancing urban hydrological and climate studies. The current study calls attention to disciplinary barriers that may prevent the two communities, urban climate and urban water management, from communicating well. The key findings are summarised below, and the corresponding recommendations are followed respectively:

(1) - Integrating various models with different foci is crucial for advancing research at the intersection of urban meteorology and hydrology. There are both challenges and opportunities in simulating the urban hydro-meteorological environment at regional and neighbourhood scales.

- WRF-Hydro coupled with urban land surface models (ULSMs), and computational fluid dynamic (CFD) models coupled with hydraulic-hydrological models show potential for regional and neighbourhood scales in urban hydro-meteorological studies.

(2) - Urban land surface models (ULSMs) predominantly adopt grid-based frameworks. However, their grid-to-grid energy, water, and momentum interactions are limited. In contrast, urban hydrological models (UHMs) establish direct links between individual simulation units and features such as sewer systems, channels, and rivers. The topography of the simulation domain significantly influences water transfer.

- Grid-to-grid linkages are key points for coupling the ULSMs and UHMs. Topography data and building information data sets should be considered while integrating the urban hydra-hydrological processes and thermal environments.

(3) - Both ULSMs and UHMs rely on atmospheric conditions as input. ULSMs excel at resolving near-surface conditions over urban areas, capturing average values for each simulation unit. Some models even provide vertical profiles within the urban canopy. Notably, ULSMs exhibit bidirectional integration capabilities with atmospheric models, whereas UHMs lack this feature.

- Gaps are shown among the parameterizations of atmospheric and terrestrial hydrological processes and the urban thermal environment. The evaluation of interaction among these is limited and needs effort.

(4) - The surface water balance in ULSMs integrates evaporation, which is expressed as latent heat flux, thereby linking the water and energy balances. Conversely, UHMs may neglect or aggregate evaporation processes. ULSMs typically include multiple subsurface soil layers with vertical water transfer, predominantly within the unsaturated zone. However, their depiction of the saturated zone is often limited. ULSMs are seen to incorporate more physical processes to solve the soil moisture budget in urban areas. Compared to ULSMs, UHMs pay more attention to saturated soil layers and lateral runoff. Notably, most ULSMs lack a comprehensive hydraulic system, with pipe capacities often set as unlimited for surface runoff.

- More attention should be paid to the subsurface, surface, and atmospheric processes and human activities. To achieve this, it is necessary to break down disciplinary barriers and create a multidisciplinary cross-communication platform.

(5) - Regarding land use types, ULSMs address the surface energy balance for most categories except water surfaces. In canyon concept-based ULSMs, intense vegetation development plays a crucial role in hydro-meteorological processes. Unfortunately, ULSMs tend to represent water surfaces inadequately. Conversely, UHMs do not explicitly solve the surface energy balance and lack detailed representations of building and land use types.

- Water bodies are one of the most popular nature-based solutions for climate and hydrological disasters. Not only from the process aspect but also from the urban planning aspect, future research is recommended to investigate the role of urban water bodies in the urban thermal environment and their link to urban hydrological conditions.

(6) - Snow-related processes receive attention in only a subset of ULSMs. UHMs generally consider snow melt as one of the inputs for the surface runoff. Even fewer models included snow-related human activities.

- Urban climate studies have traditionally concentrated on heat-related processes, often overlooking snow-related phenomena. However, with the advent of climate change and a growing emphasis on hydro-meteorology, it is crucial to broaden this focus. Not only should heat be considered, but urban snow and its interaction with human activities must also be examined from both thermal and hydrological perspectives. This shift is essential as snow processes and their impacts on urban environments become increasingly significant, especially in cold regions.

*Author contributions.* XC, AD, MCG, and RU conceptualized the review approach and designed the methodology; XC, JAvdW and MCG collected and categorized the literature; XC analyzed the data; XC visualized the results; XC wrote the paper draft; XC, JAvdW, AD, MCG, and RU reviewed and edited the paper; MCG and RU supervised the research activities; RU acquired funding and managed the project.

*Competing interests.* One of the authors is a member of the editorial board of Hydrology and Earth System Sciences. The peer-review process was guided by an independent editor, and the authors also have no other competing interests to declare.

*Acknowledgements.* This work is supported by the 4TU program HERITAGE (HEat Robustness in Relation To AGEing cities), funded by the High Tech for a Sustainable Future (HTSF) program of 4TU, the federation of the four technical universities in the Netherlands. This research was performed within the context of the TU Delft Climate Action Programme.

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
