# Peer review of "Barriers of urban hydro-meteorological simulation: a review"

_EGUsphere, 2024_

## Author Comment (AC2)

Specific Comments.

LL 6 and also LL 103-104. Please note that not all hydroclimatological events listed here will lead to "compound events". "Compound events" is used a bit loosely in the manuscript.

We acknowledge that not all hydroclimatological events listed will necessarily result in compound events. We will revise the manuscript to ensure that the term "compound events" is used more precisely and appropriately with the definition.

We will revise the manuscript:

*As our understanding deepens and the urgency to address future climate disasters grows, it becomes clear that hydrological disasters—such as floods, droughts, urban heat island, and more frequent heat waves- cannot be considered in isolation. These disasters can occur simultaneously or successively, creating compound events (Wehner, 2023; Zscheischler et al., 2020). Evaluating the compound impact of these interconnected disasters is crucial, as it provides a more comprehensive understanding than considering each disaster in isolation.*

LL 63. "...small scale heterogeneity" is also characteristic of many natural systems not only urban.

We will revise the sentence in the manuscript as below:

*Cities are characterized by dense populations and intense socio-economic activity, leading to intensive landscape modifications. Meteorology is notably intricate and heterogeneous in urban areas, and the hydrological processes differ significantly from those found in the natural environment. While small-scale heterogeneity is also characteristic of many natural systems, urban areas exhibit unique patterns and scales of heterogeneity due to human activities.*

LL 72. At this stage is not clear what a distinction between a "urban meteorology tool" and "urban land surface model" is. A U-LSMs to be an urban meteorology tool needs to be coupled with a mesoscale weather model, otherwise at the best it can explore "micrometeorology" in the canyon, not the overall city meteorology.

We will revise the phrase and the sentence to be more rigorous. To avoid ambiguity, we re-wrote these sentences in lines 72 – 73:

*To adequately simulate urban climate dynamics, several land surface parameterization schemes for urban areas have been developed in recent years, sometimes including urban land-atmosphere exchange. Urban land surface models (ULSMs) have been intensely applied to simulate urban thermal conditions in the past decades.*

LL 82-82. It might be worth referring to Jongen et al 2024, which is looking to "hydrological aspects" of these U-LSMs, already at this stage and move the later text here.

We will move the text "*Recently, Jongen et al. (2024) evaluated the water balance in 19 ULSMs from the Urban-PLUMBER project (Lipson et al., 2024), concluding that the water balance appears unclosed in 43% of the model runs. The interactions between the urban water cycle and the energy budget may not be comprehensively captured by the existing models.*" to the second paragraph of the Introduction.

And in the fifth paragraph of the Introduction section, we will revise the sentences: *Current process-based simulation tools, such as ULSMs and UHMs, have evolved independently, each driven by distinct considerations and applications. This divergence has resulted in limited overlap and interaction between the two. Consequently, accurately simulating urban hydrological and near-surface meteorological conditions—collectively referred to as hydro-meteorological conditions—simultaneously remains challenging. This makes it difficult to assess the risk of extreme events, particularly compound events, and to develop diverse strategies to address future climate change.*

LL 89-90. I think this sentence is not clear, I would suggest rephrasing it.

We will change the sentence into:

*Regarding hydrological processes in urban areas, there is no universally accepted definition of the urban water cycle. However, many texts have previously agreed on dividing the system into two main networks: modified natural pathways and supply-sewerage pathways (Lerner, 2002; Dwarakish and Ganasri, 2015).*

LL 163-164. While it is true that they represent all urban facets, many CFD approaches prescribe surface temperature of urban facets or of a subset of urban facets, which means they are not solving for energy budget (or at least not fully solving for it).

We will revise the description of the Building-resolving models to be accurate.

LL 172-173. I would suggest rewriting this sentence, it reads awkwardly.

We will rewrite the sentence into: *Studies using CFD models to explore microscale urban climate, energy balance, and wind environment are common, but rarely focus on water balance processes (Lipson et al., 2024).*

Table 1. For UT&C please see also Meili et al 2025, with further model developments.

We will incorporate this study to our review and add it into Table 1.

LL 221. How are lateral water flows solved in SUEWS?

We will add a description on how and to what extent SUEWS addresses the lateral water flow among different types.

Based on literature, in SUEWS, the surface water lateral flow within the grid cell, but among the different land use types, is simulated by the following steps:

Each surface calculates its drainage based on the water balance equations. The water distribution matrix, which specifies how water is redistributed among different surface types, will be used for the surface lateral flow process. For example, water drained from a paved surface might be partially redirected to nearby grass or soil surfaces. When the infiltration capacity is exceeded, excess water becomes surface runoff. The model calculates potential runoff for each surface type and redistributes it according to the water distribution matrix.

Besides, SUEWS accounts for both above-ground (surface runoff) and below-ground (subsurface flow) water movements. This ensures a comprehensive simulation of lateral water flow. We will include this part in section 5.4.1 and Table 7.

LL 255. Please note that a main advancement of UT&C is the capability to consider physiological and biophysical properties of vegetation, and thus being able to consider different vegetation types – at least in principle - which was not the case in any of the other models except partially VTUF-3D

Thank you for your valuable insight on the model UT&C. We will add the mentioned main advancement of the UT&C in the manuscript.

LL 261. They take "wind speed" too as input. I suggest having the complete list of six environmental variables. Pr, Ta, RH, Ws, Rsw, Ldw, some models need atmospheric pressure too.

Indeed, some models take wind speed as input forcing.  We will revise the sentences and the Figure 2.

LL 276. The second acronym should be TUF-3D, without "V".

We will revise this, thank you for checking.

LL 313 and below. Maybe using "grid cell" rather than "square" would be more aligned with the previous literature. Overall, I think this part can be shortened.

We will modify the manuscript using the "grid cell".

LL320-333. I find all this part a bit convoluted and not very clear, please consider rewriting and probably shortening, as it might not be so essential.

The paragraph is shortened and more concise:

*Figures 3c and d show the structures of the coupled mode in climate models, which simulate the entire study area in one go. These models use square grid cells of uniform size. SUEWS, similar to a complete land surface model, includes impervious land. Recently, WRF-SUEWS (Sun et al., 2023) was developed as a land surface scheme. The main difference between standalone and coupled SUEWS is the interaction of atmospheric conditions with the land surface. The coupled mode uses Urban Canopy Models (UCMs) for impervious areas, with the fraction set manually based on urbanization or land use data. The land surface model handles other landscapes like grass or bare land. Interactions between impervious and other land areas occur indirectly through air exchange. Coupled mode simulations resolve grid-to-grid atmospheric differences but limit grid-to-grid interactions. Coupled mode 2D UCMs focus on impervious land, while standalone UCMs integrate various urban canyon processes (Ryu et al., 2016; Wang, 2014; Lemonsu et al., 2012). Thus, standalone models are usually more advanced and comprehensive for urban process (Krayenhoff et al., 2020; Meyer et al., 2020).*

LL346-347. Please note that "urban block scale" and "neighborhood area" are quite vague and actually they might largely overlap, a clarification about scale might be helpful here.

We acknowledge that the terms "urban block scale" and "neighborhood area" have a significant overlap. We will change the description enhancing the precision and clarity.

Consequently, UCM models are more widely utilized for studying regional impacts at the hundred-meter scale, and high-resolution CFD models are primarily used for detailed research at the meter scale.

LL 350. "present" rather than "utilise"

We will revise this.

LL 402-403. Please note that data availability might be an issue when defining branching and drainage structures in many urban environments, as sewage system locations are often poorly known in old cities or not available in modern cities. This can also be discussed later.

Indeed, we totally agree that the availability of data is an essential issue. We will include this in the challenge and future development section of the manuscript.

In Section 6 we have a discussion on this:

*However, both 3D ULSMs and URBS hydrological models require high-resolution input data, such as urban morphology, topography, and sewer systems. This need for detailed urban data limits the application of URBS compared to models like SWMM. As data*

*collection and measurement methods for urban morphology and infrastructure improve, high-resolution simulations of urban microclimates and hydrological processes at the neighborhood scale may become feasible, enabling comprehensive thermal and hydro-hydraulic interactions*.

LL 435. Are you referring to water surface as "urban lakes" or "ponding water in various surfaces" the latter is likely solved in most U-LSMs.

We are referring to urban water surfaces, including urban lakes. We will revise the sentence to be clearer. Our review indicates that most Urban Land Surface Models (U-LSMs) do not incorporate these urban water surfaces. Specifically, only the SUEWS, TARGET, and Solene-Microclimat models resolve the energy balance above water surfaces. In most Urban Canopy Models (UCMs), only canyon elements (streets and buildings) and canyon vegetation are represented, meaning water surfaces are not addressed within an urban grid. In coupled models, where ULSMs are integrated with regional climate models, all water surfaces are simulated solely by the Land Surface Model (LSM). Our study highlights the need to consider how urban water surfaces, which can differ significantly from natural landscapes, influence the urban micro-mesoscale climate.

LL 458. Please note that the shortwave and longwave radiation budget is different and not only sky view factors but view factors among different urban facets are required, at least for longwave exchange, it is also not true that all U-LSMs do not compute multiple reflections.

We will adjust the sentence.

LL 459. "radiosity method" is unclear.

Thanks for your comment. The phrase "radiosity method" is directly adopted from the literature on the Solene-Microclimat model. We have an explanation of how this method works: "*This method involves two critical aspects: 1) geometric form factors between all facets of the built surface and the sky vault are calculated using the contour integral technique; 2) radiative properties (reflection, transmission, and absorption) are considered for all surfaces within the scene*." We will revise this part of the manuscript and make the description of this method clearer.

Table 5. Maybe it could be interesting subdividing between models using a multi-layer vs two- or three-layers approach for solving the urban canyon, as more than one layer is not lumped anymore.

We agree that more than one layer is not lumped. In Table 5, if the model can solve more than one layer value within the canyon, we marked it as P (profile). We double-checked all the models to what extent they solve the hydro-meteorological conditions and distinguish the lumped (single layer), two layers and multi-layer (three and above) models. There are

only single and multi-layer models in our selected models. Thus, we keep the Lump (L) and Profile (P) for the table.

LL 495. "area" rather than "ares".

We will correct this.

LL 537. Urban Hydrological Models not "Urban Heat Models", which does not mean anything.

We will revised this, thank you for checking.

LL 598. Applying the 1d Richards equation allows for the solution of infiltration, but it is much more than an infiltration method, as it allows the solution of variably saturated flow in multiple soil layers.

Indeed, we agree with the comment. In Section 5.4.1, we mentioned that the vertical moisture transport in multiple soil layers is solved by the 1D Richards equation in SLUCM, TEB, UT&C and VUCM.

LL 604-605. The description of the "Horton method" is wrong. It is the other way around, infiltration decreases exponentially up to an asymptotic constant value at large times, which is typically the saturated hydraulic conductivity.

We double-checked the Horton method, and we agree with your comments. We will revise the manuscript:

*The Horton method is an empirical model. It assumes that infiltration decreases exponentially over time until it reaches a constant value, typically the saturated hydraulic conductivity.*

LL 615. Please note that on most impervious surfaces (e.g., roofs, paved streets), the saturated hydraulic conductivity will be fundamentally zero, so why one should compute infiltration? This is not clear to me.

You are correct that infiltration over impervious surfaces is negligible, which is consistent with the approach taken by most ULSMs. Our intention is not to suggest that models should compute infiltration over impervious land. Rather, we are highlighting that some models set a constant rate for infiltration over impervious surfaces, while others do not. This is a summary statement. We will make it clearer in the manuscript and propose the following sentences:

*Generally, if a model includes the water balance, it considers depression storage and infiltration. Based on the above summary, depression storage over impervious land areas*

*can be solved relatively easily. The infiltration process over impervious land is not taken into consideration because the saturated hydraulic conductivity of these surfaces (e.g., roofs, paved streets) is fundamentally zero.*

LL623. Do you mean Table 7?

Yes, we will revise it.

LL 638. In VTUF-3D and UT&C transpiration is indeed simulated using an explicit solution of stomatal conductance, which is a function of photosynthesis, so plant photosynthesis needs to be solved.

Thanks for your comment. We will expand our description of the process of solving the transpiration in VTUF-3D and UT&C.

LL646-656. I think here, there is a bit of confusion around Penman-Monteith equation method. If a model solves the energy budget (i.e., latent heat) by solving for surface temperature/s, there is no need to use Penman-Monteith equation, which is actually a simplification of the overall energy budget and it does not guarantee energy budget closure. I think which models use Penman-Monteith and which aren't should be better described and specified.

We will summarise this and will make it clear in the manuscript. The paragraph starts from LL 646 and will be revised.

There are several groups of methods to calculate the evaporation:

1) SUEWS (Ward et al., 2016) and WEP (Jia et al., 2001): calculated based on the Penman-Monteith equation. The models first obtains the net radiation, equal to the sum of the heat fluxes, including sensible, latent and ground heat fluxes. The storage heat flux and anthropogenic heat flux will be calculated, and then the latent heat flux will be calculated based on the Penman-Monteith equation. The energy balance closure will be solved by setting the residual as the sensible heat flux.
2) VUCM, ASLUCM, TEB, UT&C, VCWG (Lee, 2011; Meili et al., 2020; Masson, 1999; Wang et al., 2013): solve the evaporation using the resistance method. This method directly calculates the moisture heat flux based on the saturation value of specific humidity at a given surface temperature. These models directly solve the energy balance and the surface temperature.
3) URB (Berthier et al., 2004): The evaporation flux is assumed to be proportional to the potential evapotranspiration and water storage, while the potential evapotranspiration is the model's input forcing.
4) SWMM (Rossman,2017): three simplified ways, including setting as a single constant value, getting from the input data, and computing the daily values from the daily temperatures in an external climate file

LL 692-694. Does this refer to canopy or ground vegetation? This is not clear.

We double-checked the literature Lee and Park, 2008. The vegetation referred to in the study is canopy vegetation. We will make it clear in the manuscript.

LL723. The end of the sentence is not clear, which empirical models? Which calibration?

We will revise this part of the manuscript with more details as follow:

*Soil conductivity and saturated soil conductivity in urban hydrological models are consistent with those represented in most Urban Land Surface Models (ULSMs). These parameters can be calibrated in the field using infiltrometers and laboratory measurements, such as the constant head and falling head methods (Gupta et al., 2021). Additionally, they can be estimated using empirical models based on soil type, moisture content, and temperature, such as Pedotransfer Functions (Van Looy et al., 2017). The accuracy of these estimates can be further enhanced through machine learning techniques, including artificial neural networks and random forests (Jian et al., 2021).*

LL770-772. I am not sure I understand the question here and especially why it is formulated as a question. It is mostly a parameterization issue how to deal with these processes.

Reply to the comments LL 770 – 772: We will revise the sentence:

*As stand-alone ULSMs increasingly incorporate natural land surfaces and interaction processes, it becomes imperative for regional atmosphere-land models and ULSMs to systematically address the radiation, aerodynamic, and dynamic interactions between natural landscapes and anthropogenically influenced landscapes, particularly water surfaces and built-up areas.*

LL 778. I am not sure I understand why ones would like to have two-ways feedback from the hydraulic part to the atmosphere, being the flood response mostly a very fast process and occurring largely in impermeable channels and drains.

We understand your perspective on the rapid nature of flood response and the role of impermeable channels and drains. Indeed, the atmospheric simulation output can be the input for the urban flooding forecast. However, we believe there are several reasons why linking atmospheric models, land surface models, and hydrological-hydraulic models can be beneficial: 1) Two-way feedback allows for more precise monitoring and control of hydraulic systems, which can improve the accuracy of flood predictions and responses. Currently, the hydraulic model takes a single grid precipitation data as the input. However, with a deeper understanding of the urban impact on precipitation events, the spatial variability can be quite large, even within a catchment, depending on its size. Thus, higher-resolution input data for hydraulic systems can provide better performance. 2) Two-way

feedback can help manage waterlogging more effectively by providing real-time data on soil saturation and atmospheric conditions. This can lead to better predictions and responses to waterlogging events. 3) Two-way feedback can improve the simulation of the urban micro-climate during and after extreme precipitation events that may lead to urban flooding. This can also optimize drainage and reduce the negative impacts on local urban ecosystems.

To be clearer, more accurate and specific, as also suggested by reviewer 3, we will revise our manuscript for Section 6, Challenges and future developments, to include a more detailed explanation of the necessary model coupling. The revised Section 6 explicitly frames future developments around specific urban hydroclimate challenges, including urban thermal environment adaptation, urban flood forecasting, and compound extreme event analysis. The necessity and significance of model coupling strategies and modelling approaches are addressed for each specific topic.

LL 875. As the authors know, total evaporation and latent heat flux is the same thing, just different units, the sentence is awkward.

Yes, we will revise the sentence:

*The surface water balance in ULSMs integrates evaporation, which is expressed as latent heat flux, thereby linking the water and energy balances. Conversely, UHMs may neglect or aggregate evaporation processes. ULSMs typically include multiple subsurface soil layers with vertical water transfer, predominantly within the unsaturated zone. However, their description of the saturated zone is often limited.*

References

Meili, N., Zheng, X., Takane, Y., Nakajima, K., Yamaguchi, K., Chi, D., Zhu, Y., Wang, J., Qiu, Y., Paschalis, A. and Manoli, G., 2025. Modeling the effect of trees on energy demand for indoor cooling and dehumidification across cities and climates. Journal of Advances in Modeling Earth Systems, 17(3), p.e2024MS004590.

---

## Author Comment (AC3)

Major Comment 1: The paper would benefit from a conceptual schematic and stronger directory-style guidance, ideally integrated into Section 2.

While the manuscript does mention its rationale for selecting representative and newly developed models and those at the climatology-hydrology interface, it currently lacks an overarching conceptual framework that visually summarises the overall structure of the review. Including a schematic diagram early in Section 2, clearly illustrating how different urban processes (such as radiation, evapotranspiration, runoff, and soil moisture) relate to the various model classes reviewed (bulk, SL-UCM, ML-UCM, hydrology-focused models) and coupling strategies, would provide valuable directory-like guidance. Such a visual roadmap would help readers, especially those less familiar with the fxield, to better navigate and contextualise the rich and diverse content of subsequent sections.

We will incorporate this conceptual framework of the overall structure to provide readers with a clearer understanding of the connections and context of the models reviewed. We will incorporate the framework at the end of Section 1 while explaining the following structure and contents.

[Figure]

**Figure 1.** The overall schematic structure of the review.

Major Comment 2: The future directions section should be more closely tied to specific scientific tasks and supported by clear technical mechanisms.

Currently, the call for integration and collaboration in Section 6 is valuable but somewhat general. To strengthen its practical impact, the authors could explicitly frame future development around specific urban hydroclimate challenges (e.g. urban flood forecasting, heat mitigation, compound event analysis), clarifying the necessary coupling strategies and modelling approaches.

Two concrete, actionable suggestions could help operationalise this vision:

Develop a common modelling protocol—similar to the NetCDF CF conventions widely used in climate sciences—which is currently missing for urban hydroclimate modelling. Such a protocol could standardise inputs, outputs, metadata, and resolution criteria to greatly facilitate interoperability and cross-model comparisons. Initial steps toward this goal have already been taken within the Urban-PLUMBER initiative, but these efforts should be expanded and formalised into more broadly accepted community practices.

Establish a practical technical framework to overcome barriers in collaborative model development. Leveraging open-source collaboration platforms like GitHub could foster transparency, modular development, and community engagement. Additionally, considering WRF's wide adoption and flexibility, it might serve effectively as a common base framework to host and test integrated urban hydroclimate components.

These steps would provide concrete pathways to improve the current fragmented landscape, enabling more coherent and coordinated model advancements.

We concur that focusing future development on specific urban hydroclimate challenges will significantly enhance the practical impact of our call for integration and collaboration.

We will revise our manuscript for Section 6, Challenges and future developments, to include a more detailed explanation of the necessary model coupling. The first three parts of the revised Section 6 explicitly frame future developments around specific urban hydroclimatic challenges, including urban thermal environment adaptation, urban flooding forecasting, and compound extreme events analysis. The necessity and significance of model coupling strategies and modelling approaches are addressed for each specific topic:

*6.1 Urban thermal environment adaptation*

*The thermal environment depends not only on temperature, but also on humidity. Thus, accurately estimating these two terms is essential to take into account. Two directions of efforts have been made in the last decades on ULSMs.*

*First, ULSMs are widely implemented in regional climate models, as mentioned in Section 3 and Figure 2. Coupling the atmospheric models with land surface models, including ULSMs, supports studies of land-atmosphere interactions, specifically the impact of urban*

*land on this interaction. Recent studies have further employed physically realistic, coupled atmosphere-land surface-subsurface models (Wagner et al., 2016; Fersch et al., 2020; Kim et al., 2021). These coupling simulations aim to evaluate the intricate interplay between terrestrial and atmospheric processes, shedding light on their influence on hydro-meteorological phenomena. By coupling the WRF-SLUCM and a land surface-subsurface model (ParFlow), Talebpour et al. (2021) emphasized the critical need to account for terrestrial hydrology, particularly in urban areas where diverse development patterns introduce additional complexities to coupled atmosphere-groundwater interactions. The coupling between the WRF-SLUCM and ParFlow can overcome some of the limitations of representing the saturated soil layer and groundwater in the ULSMs found in the current study.*

*Second, the ULSMs are further developed, accounting for complex physical processes, notably hydrological processes and the representation of vegetation, to support studies on the interface of meteorology and hydrology. In general, the development of ULSMs initially focused on stand-alone models, which have subsequently been implemented in coupled versions with mesoscale climate models. Therefore, stand-alone versions are usually more advanced in physical complexity than their coupled versions. This is the case for, e.g., ASLUM (Wang et al., 2013) and WRF-SLUCM (Yang et al., 2015), as well as the SUEWS and WRF-SUEWS (Sun et al., 2023; Sun and Grimmond, 2019). However, the hydrological module of BEP was first implemented in a coupled version by Yu et al. (2022) (WRF-BEP) instead of as a stand-alone version. In the version of WRF-BEP, depression storage over impervious land, evaporation over urban green surfaces, and pipe system parameterizations based on Manning's formula are included (Yu et al., 2022).*

*However, urban land surface models (ULSMs) face challenges in simulating natural land surfaces, including water and vegetation, within urban areas. Current mesoscale land-atmosphere simulations focus on impervious urban areas, but urban regions also contain vegetation and lakes. These natural patches are crucial for simulating the urban thermal environment, affecting energy balance and humidity levels. Based on our study, stand-alone ULSMs have started incorporating natural land surfaces, including vegetated land and trees, but often exclude urban lakes and ponds. It is recommended to include urban water surfaces in ULSMs to address interactions between natural and built-up landscapes. Using large eddy simulation with mesoscale climate models can improve spatial resolution and capture small natural landscapes within urban areas. Adjustments are needed for simulating urban lakes due to differences from natural lakes. The Local Climate Zone scheme should also consider urban water landscapes in its classifications.*

*6.2 Urban flooding forecasting*

*Accurate precipitation data and coupling of hydrological models with hydraulic models contribute to accurately forecasting urban flooding and inundation. Atmospheric models have been coupled one-way with urban hydraulic models. Coupling enables more precise monitoring and control of hydraulic systems, thereby enhancing the accuracy of flood predictions and responses. Traditionally, hydraulic models utilize single-grid precipitation data as input; however, understanding the urban impact on precipitation events reveals significant spatial variability within a catchment (Cristiano et al., 2017). Consequently, higher-resolution input data from atmospheric models can improve hydraulic system performance. Furthermore, coupling allows to more effectively manage inundation by providing real-time data on soil saturation and atmospheric conditions, leading to improved predictions and responses to inundation events. For example, recently Gu et al. (2022) coupled the WRF with SWMM to provide precipitation input for their hydrologic-hydraulic simulation.*

*Urban land modifies both atmospheric and terrestrial hydrological processes. By leveraging the two-way coupling between land and atmospheric models, researchers can explore the effects of urbanization on atmospheric and terrestrial hydrological processes, respectively. For instance, an emerging study area involves investigating urban-induced convective precipitation events (Wang et al., 2021b; Yang et al., 2024). Accurately simulating high-resolution rainfall is challenging due to its spatial variability and randomness. There is another group of existing studies using hydrological models to investigate the impact of urbanization on terrestrial-hydrological processes, such as runoff and infiltration (Oudin et al., 2018; Locatelli et al., 2017; Yoo et al., 2021). However, these investigations often overlook atmospheric interactions, concentrating solely on land surface and subsurface hydrological processes. This alignment with the findings in the current analysis underscores the complexity of terrestrial-hydrological processes, which exhibit strong coupling with atmospheric dynamics.*

*To enhance urban flooding forecasts, it is crucial to integrate atmospheric, hydrological, and hydraulic models effectively. Accurate precipitation forecasts require a well-represented land surface in a two-way coupled land-atmospheric simulation.  At the same time, the hydrological model WRF-Hydro shows potential for coupling with a land-surface model and an atmospheric model (Gochis et al., 2018). This can provide an opportunity to study urban impacts on terrestrial-hydrological processes and their subsequent effects on atmospheric-hydrological interactions simultaneously. In the end, hydraulic models take both outputs from the atmospheric and hydrological models to better simulate urban flooding and inundation.*

*6.3 Compound extreme events analysis*

*Extreme weather events like heavy rainfall, floods, droughts, and heatwaves can lead to compound disasters. Urban water bodies and temporary water accumulations from rainfall, along with drought conditions, significantly impact the urban thermal environment by altering energy and humidity levels (Hao et al., 2023). Furthermore, urban heat islands can affect convective rainfall amount and distribution, and influence land hydrological processes. Understanding the interconnections of these events is crucial for assessing climate disaster risks. This involves studying land-atmosphere interactions and coupling thermal and hydro-hydraulic processes to improve urban micro-climate simulations, optimize drainage, and mitigate adverse effects on ecosystems.*

*Coupling and integration are mainly applied to land surface and atmospheric models. One key challenge in fully implementing hydraulic and terrestrial-hydrological processes with thermal processes is the need for interaction between simulation units, such as grid cells in ULSMs like Multi-Hydro and WEP, and sub-catchments in SWMM and UHE in URBS. WRF-Hydro, coupled with a land surface model, includes terrain routing modules for 2D overland flow and soil moisture links (Gochis et al., 2018). It also features channel and reservoir routing modules and can be two-way coupled with the atmospheric model. WRF-Hydro can be coupled with SMWW for urban hydrology and drainage (Son et al., 2023). However, its development and application in urban areas and the interface between urban thermal and hydrological environments are limited (Fersch et al., 2020).*

*Advancements in computing systems have led to the use of CFD models, traditionally for urban wind analysis, in simulating urban thermal environments. Despite limitations in 3D ULSMs for hydrological cycles, they show potential for urban hydro-thermal environments when integrated with distributed hydrological models (Robineau et al., 2022). Both 3D ULSMs and URBS hydrological models need high-resolution input data, such as urban morphology, topography, and sewer systems, limiting URBS compared to SWMM. Improved data collection methods may enable high-resolution simulations of urban microclimates and hydrological processes at the neighborhood scale, allowing comprehensive thermal and hydro-hydraulic interactions. CFD models are advantageous at neighborhood and microscale levels for studying climate-hydrological mitigation measures. Further development is needed to integrate physical processes with high-resolution hydraulic-hydrological models, considering data availability and computational resources.*

In the last part of Section 6, we discuss the actionable suggestions to enhance the collaboration within and across disciplines:

*6.4 Collaboration within and across disciplines*

[revised manuscript text omitted]

L213:  The authors state that "Wang et al. (2016) added an irrigation scheme to the model" (referring to SUEWS). This attribution is incorrect. The irrigation functionality was first introduced in Järvi et al. (2011), which documented the initial release of the Surface Urban Energy and Water Balance Scheme (SUEWS). The cited Wang et al. (2016) paper does not

pertain to SUEWS development and focuses instead on vegetation cooling in desert cities. I suggest correcting this reference to reflect the accurate model development history.

Reply to comment on L213:

We will revise this part of the manuscript.

---

## Author Response (AR1)

Contents:

Comments from Zhihua Wang (Reviewer 1):

This paper presents a comprehensive review of the state-of-the-art urban climate and urban hydrological modeling. The review is timely and of great interest to the urban study community, especially because urban hydrological modeling has been lagging behind practices for sustainable urban development. While some technical details need to be further clarified (see my specific comments below), overall, the paper is technically sound and well written, and I thoroughly enjoyed reading the manuscript. I therefore recommend the paper to be accepted for publication after the following comments to be adequately addressed.

Dear Zhihua,

Thank you very much for your thorough and thoughtful review of our manuscript. We are happy to hear that you found our review of urban climate and urban hydrological modeling to be comprehensive and timely.

We carefully addressed each of your specific comments to ensure that the manuscript meets the higher standards of clarity and rigour. Please see the response to specific comments below.

Best regards,

Xuan and all the co-authors

**Specific comments**

1. Lines 112: on the compound urban climate mitigation mechanisms, the following study provides a mathematical formalism that may worth considering:

Wang, Z.H. (2021). Compound environmental impact of urban mitigation strategies: Co-benefits, trade-offs, and unintended consequence. Sustainable Cities and Society, 75, 103284. https://doi.org/10.1016/j.scs.2021.103284

We incorporated these relevant insights to enhance our revised manuscript on line 121. We added the sentence:

*Wang (2021) proposed a mathematical framework for unified evaluation of compound impacts emphasizing that compound urban climate mitigation should be evaluated in a comprehensive way.*

2. Line 160-161: "However, the multilayer models solve the vertical profiles of the atmospheric conditions", the phrase "atmospheric conditions" is too general to describe the multilayer UCM, it is more precise to use "canopy-layer flows and momentum transport".

We revised the 'atmospheric conditions' into a more detailed description of 'canopy-layer flows and momentum transport':

*The single-layer models treat the urban canyon as a homogeneous area. However, the multilayer models solve the vertical profiles of the canopy-layer flows and momentum transport.*

3. Table 1: under SLUCM, the citation Wang et al., 2021 should be referring to

Wang, C., Wang, Z.H., & Ryu, Y.H. (2021). A single-layer urban canopy model with transmissive radiation exchange between trees and street canyons. Building and Environment, 191, 107593. https://doi.org/10.1016/j.buildenv.2021.107593

which is not included in the reference list.

We revised the reference in Table 1 and add it to the reference list.

4. Figure 1: I wonder if it is necessary to separately indicate the surface temperature and net radiation for heterogeneous landuse. For by the same token, sensible (and latent) heat fluxes for these facets are also different and should also be separately indicated.

In addition, there is a downwelling $R_{net}$ and 7 upwelling $R_{nets}$, while the net radiation from the urban canopy should be the combination of them, and none of the individual components can be called $R_{net}$ I'd suggestion to keep the separate representation of surface temperatures as it is, but combine the radiation into a single $R_{net}$ (or with a downwelling radiation as $R_{down}$ and a upwelling component as $R_{up}$).

We agree with the reviewer. We deleted the surface temperature at each type of surface, but kept the evaporation-LE based on Figure 1 in Järvi et al. (2014). The net all-radiation and lumped sensible heat and latent heat fluxes are calculated for the whole grid. And indeed, the $R_{net}$ should be the combination.

We updated Figure 1 as you suggested. Please see the revised figure below.

[Figure]

5. Lines 226-229, "The single-layer urban canopy model developed by Kusaka et al. (2001) (SLUCM) is very similar to TEB. The only differences are that the SLUCM in this version (Kusaka et al., 2001) includes the canyon orientation and diurnal change of solar azimuth angle, and the surface consists of several canyons with different orientations." This statement is incomplete. In fact, Kusaka's SLUCM, as implemented into WRF, contains a different parameterization scheme of radiation by discretizing the canyon facet and computing radiation on individual gridcells, whereas TEB uses the analytical formulae for in-canyon view factors. For simple rectangular canyons with only walls and roads (and short vegetation), Kusaka's radiation scheme is a setback to the analytical formulation, but it opens the possibility to include radiative exchange by roughness elements presented in street canyons such as trees/blocks/vehicles.

We double-checked Kusaka's SLUCM, ASLUM and TEB, especially the radiation parameterization scheme, as you mentioned, and revised the manuscript as follows:

*Masson (2000) developed a single-layer urban canopy model called the Town Energy Budget (TEB) scheme. Similarly, Kusaka (2001) introduced the Single-Layer Urban Canopy Model, which is integrated into the Weather Research and Forecasting model (WRF-SLUCM). Unlike TEB, which uses analytical formulas for in-canyon view factors, WRF-SLUCM employs a distinct radiation parameterization scheme by discretizing canyon facets and calculating radiation for individual grid cells. Although Kusaka's scheme may be*

*less efficient for simple rectangular canyons with walls, roads, and short vegetation, it has the advantage of accounting for radiative exchanges involving roughness elements such as trees, blocks, and vehicles within street canyons.*

6. Table 3, I don't really understand how the temporal resolution of different ULSMs is determined. If the temporal resolution refers to the time intervals/steps used to solve the parameterization scheme, it varies widely depending on the discretizing (forward- or central-in-time finite difference) schemes, running platforms (offline or imbedded in regional climate models such as WRF), and applications. The time steps used to solve parameterization schemes can be as small as 1s (e.g. WRF-UCM for it used both spatial-temporal discretization for land-atmosphere interactions), or as large as 30 min. If the temporal resolution refers to the time scale for sampling the output, it is a rather arbitrary choice of the users. For instance, output of WRF-UCM is often sampled in hourly scale, like what is indicated in the table, but it can also be sampled at 3-hourly or 6-hrouly intervals for longterm (monthly to annual) simulations, but it can also be, in theory, sampled at 1s interval. My understanding is that the temporal resolution of all UCMs have no essential difference as their parameterization schemes represented by partial differential equations of land surface processes are all similar. The spatial scale for them does vary for SLUCM resolves the physical structure of the canyon, and building-resolving models has to resolve individual buildings, while slab models represent the aggregated urban landscapes.

In the previous manuscript, we tried to summarize the temporal resolution of the model outputs, as you said, at the hourly scale for WRF-UCM. Our review indicated that the analysis of model outputs for urban climate applications is typically conducted on a minute-to-hourly basis. However, we acknowledge your comment that the temporal resolution of UCMs should be determined by the needs of the users rather than the simulation capabilities of the model. It can be monthly or yearly, and theoretically, it can be seconds. Therefore, we modified the table and we replaced the previous table with below revised one. The temporal resolution indicates the time scale at which model output is sampled.

We also added the following description addressing this issue:

*The temporal resolution refers to the time scale for sampling the model output. Although the analysis of the model output is a rather arbitrary choice for users, the table shows the general timescale at which the model output is analyzed. The temporal resolution of the 2D ULSMs is above half an hour and can reach the minute level for the 3D ULSMs.*

**Table 3.** Information of included models. The information is based on the literature listed in Table 1 and Table 2 and also on Salvadore et al. (2015). The column of Temporal Resolution is marked by *, indicating the time scale at which model output is sampled.

| Shortname | Type | Simulation Unit | Spatial Resolution | Temporal Resolution* |
|---|---|---|---|---|
| SUEWS (WRF-SUEWS) | Bulk (2-Tile) | Grid cell | 100 m - 5 km | min - hr |
| VUCM (WRF-VUCM) | Canopy-Single layer | Grid cell | 100 m - 5 km | min - hr |
| ASLUM (WRF-SLUCM) | Canopy-Single layer | Grid cell | 100 m - 5 km | min - hr |
| TEB (WRF-TEB) | Canopy-Single layer | Grid cell | 100 m - 5 km | min - hr |
| TARGET | Canopy-Single layer | Grid cell | 100 m - 1 km | min - hr |
| UT&C | Canopy-Single layer | Grid cell | 100 m - 1 km | min - hr |
| VCWG | Canopy-Multi layer | Grid cell | 100 m - 1 km | min - hr |
| BEP (WRF-BEP) | Canopy-Multi layer | Grid cell | 100 m - 5 km | min - hr |
| VTUF-3D | 3D Building resolved | Mesh cell | 1 m - 10 m | sec - hr |
| Solene-Microclimat model | 3D Building resolved | Mesh cell | 1 m - 10 m | sec - hr |
| SWMM | 1D Hydraulic | Sub-catchment | 100 m - | min |
| Multi-Hydro model | 1D Hydraulic-hydrological | Grid cell | 100 m - | min |
| URBS | 1D Hydraulic-hydrological | UHE | 10 m - 1 km | min |
| WEP | 2D Hydraulic-hydrological | Grid cell | 100 m - 5 km | min - hr |

7. Table 5: SLUMC should be SLUCM (same typo in Tables 6 and 7). Also, I am concerned about the naming of the urban canopy models. The discussion of the single-layer urban canopy models in this paper is largely based TEB, Kusaka's UCM implemented in WRF, and the Arizona Single Layer Urban canopy Model (ASLUM, the name is used in Wang et al., 2021, Lipson et al., 2024, and Jongen et al., 2024). Yet TEB is separately discussed in this table, and SLUCM seemingly groups ASLUM and Kusaka's model. Given the fact that Kusaka's single-layer model is not further developed in a separate line, the representative SLUCM should be more properly named after ASLUM, as the latter is a coherent family of models developed by the same group of model developers in a continuous manner (Wang et al., 2013; Wang, 2014; Yang et al., 2015; Ryu et al., 2016; Wang et al., 2024).

We agree and modified the references to SLUCM to ASLUM throughout the whole manuscript. We double-checked the typos and revise them.

In the revised manuscript, we added a description in Section 3.2 "Urban canopy models", to distinguish between the Kusaka's SLUCM and ASLUM: "*ASLUM was developed following Kusaka's SLUCM and has been updated to the present (Wang et al., 2013)*".

8. Section 6: this part presents some thought-provoking questions that need to be pursued in future development of urban climate and urban environment modeling in depth. I would suggest the authors also include a brief discussion of the potential of AI and machine learning application in the field, given that these tools are increasingly adopted and some promising results generated from pioneering work in this field.

Your suggestion to include a discussion on the potential applications of AI and machine learning in this field is highly valuable.  Although our review primarily focuses on processbased models, the benefits of AI techniques for further model development and their potential to influence the scientific direction are indeed worthy of discussion. We incorporated this topic to enrich our manuscript and provide a more comprehensive overview of the future directions in this field.

In Section 6.4 "Collaboration within and across disciplines", we added one paragraph to discuss how AI and ML can work with physics-based models, contributing to the simulation of urban hydrometeorology, and encourage its future application:

*Building on the importance of interdisciplinary collaboration highlighted in our discussion, artificial intelligence (AI) and machine learning offer transformative potential for urban climate and environment modeling. These technologies can enhance predictive accuracy, identify complex patterns, and optimize model parameters, thereby complementing the collaborative efforts in atmospheric science, hydrology, and hydraulics. It has been shown that machine learning models and statistical models can work together with the physics-based models while being applied to different urban adaptation strategies under climate change (Li et al., 2022; Aliabadi et al., 2023). To study the urban heat, machine learning is applied for downscaling by generating high-resolution data from lower-resolution results from physics-based simulations, reducing temperature errors, and lowering computational costs (Wu et al., 2021b). Forecasting can benefit from AI and machine learning by working with numerical simulations and measurement data to enhance data accuracy and model performance for climate disasters (Luo et al., 2022). He et al. (2023) introduce a hybrid data assimilation and machine learning framework integrated into the WRF, which optimizes surface soil and vegetation conditions to improve regional climate simulations. To fully leverage these advancements, urban climatologists and hydrologists should also engage in discussions with scientists who specialize in AI applications. These interdisciplinary dialogues are crucial for integrating AI-driven insights into urban geoscience, aligning with our findings and recommendations on the necessity of interdisciplinary cooperation to address climate and natural disasters effectively.*

Comments from Reviewer 2:

The manuscript reviews the main features of (1) urban land surface models (U-LSMs) mostly used for studying urban canyon microclimate and in urban climate studies and (2) urban hydrological models (U-HMs), mostly used in urban flood analysis. By comparison these two families of models, the article shows how distant the two modeling communities are, while theoretically approaching a problem – the solution of urban energy-water-carbon fluxes – that has a lot of common elements. The authors advocate for a better integration of the two modeling approaches with better communication across communities and better integration of hydrological processes and land-atmosphere interactions.

While it is fine to develop a model for a specific purpose, I overall agree with the call made in this manuscript and the importance to review for different communities (I suppose many scientists are not aware of the full landscape of models) the differences and similarities in the various modeling approaches – with the hope that soon rather than later models across the two fields could be developed. Based on my knowledge, the article is also mostly accurate in presenting the different modeling components and I have mostly minor comments listed below. The only broader comment is that the article is currently very long, and I suppose it will benefit by some shortening, even though admittedly I could not easily identify parts that can be trimmed, even though I give a few suggestions below.

Dear Reviewer:

Thank you for your thorough and insightful review of our manuscript. We greatly appreciate your positive feedback and your agreement with the call for better integration and communication across the urban land surface and hydrological modeling communities.

We understand your concern regarding the length of the manuscript. While we believe that all the content addressed is necessary to provide a comprehensive review of the different modelling approaches, we have shortened Section 3.4, mainly the third paragraph, based on your specific comments.

Please see the response for each specific comment below.

Best regards,

Xuan and all the co-authors

 Please note that not all hydroclimatological events listed here will lead to "compound events". "Compound events" is used a bit loosely in the manuscript.

We acknowledge that not all hydroclimatological events listed will necessarily result in compound events. We revised the manuscript to ensure that the term "compound events" is used more precisely and appropriately in the definition.

We revised the manuscript to define the compound events:

*As our understanding deepens and the urgency to address future climate disasters grows, it becomes clear that hydrological disasters—such as floods, droughts, urban heat island, and more frequent heat waves cannot be considered in isolation. These disasters can occur simultaneously or successively, creating compound events (Wehner, 2023; Zscheischler et al., 2020). Evaluating the compound impact of these interconnected disasters is crucial, as it provides a more comprehensive understanding than considering each disaster in isolation.*

 "...small scale heterogeneity" is also characteristic of many natural systems not only urban.

We revised the sentence in the manuscript as below:

*Cities are characterized by dense populations and intense socio-economic activity, leading to intensive landscape modifications. Meteorology is notably intricate and heterogeneous in urban areas, and the hydrological processes differ significantly from those found in the natural environment. Although small-scale heterogeneity is common in many natural systems, urban areas display distinct patterns and scales of heterogeneity shaped by human activities, which in turn have a direct impact on people.*

 At this stage is not clear what a distinction between a "urban meteorology tool" and "urban land surface model" is. A U-LSMs to be an urban meteorology tool needs to be coupled with a mesoscale weather model, otherwise at the best it can explore "micrometeorology" in the canyon, not the overall city meteorology.

We revised the phrase and the sentence to be more rigorous. To avoid ambiguity, we re-wrote these sentences in lines:

*To adequately simulate urban climate dynamics, several land surface parameterization schemes for urban areas have been developed in recent years, sometimes including urban land-atmosphere exchange. Urban land surface models (ULSMs) have been intensely applied to simulate urban thermal conditions in the past decades.*

LL 82-82. It might be worth referring to Jongen et al 2024, which is looking to "hydrological aspects" of these U-LSMs, already at this stage and move the later text here.

We moved the text "*Recently, Jongen et al. (2024) evaluated the water balance in 19 ULSMs from the Urban-PLUMBER project (Lipson et al., 2024), concluding that the water balance appears unclosed in 43% of the model runs. The interactions between the urban water cycle and the energy budget may not be comprehensively captured by the existing models.*" to the second paragraph of the Introduction.

And in the 5th paragraph of the Introduction section, we revised the sentences: *Current process-based simulation tools, such as ULSMs and UHMs, have evolved independently, each driven by distinct considerations and applications. This divergence has resulted in limited overlap and interaction between the two. Consequently, accurately simulating urban hydrological and near-surface meteorological conditions—collectively referred to as hydro-meteorological conditions—simultaneously remains challenging. This makes it difficult to assess the risk of extreme events, particularly compound events, and to develop diverse strategies to address future climate change.*

LL 89-90. I think this sentence is not clear, I would suggest rephrasing it.

We changed the sentence into:

*Regarding hydrological processes in urban areas, there is no universally accepted definition of the urban water cycle. However, many texts have previously agreed on dividing the system into two main networks: modified natural pathways and supply-sewerage pathways (Lerner, 2002; Dwarakish and Ganasri, 2015).*

LL 163-164. While it is true that they represent all urban facets, many CFD approaches prescribe surface temperature of urban facets or of a subset of urban facets, which means they are not solving for energy budget (or at least not fully solving for it).

We revised the description of the Building-resolving models to be accurate.

The description changed into:

*Building-Resolving Models: These models normally employ computational fluid dynamics (CFD) to accurately simulate thermal and airflow conditions with 3D information of the buildings and heterogeneous urban landscape. Thus, all the urban facets are represented.*

LL 172-173. I would suggest rewriting this sentence, it reads awkwardly.

We rewrote the sentence into: *Studies using CFD models to explore microscale urban climate, energy balance, and wind environment are common, but rarely focus on water balance processes (Lipson et al., 2024).*

Table 1. For UT&C please see also Meili et al 2025, with further model developments.

We incorporated this study to our review and added it into Table 1.

LL 221. How are lateral water flows solved in SUEWS?

We added a description on how and to what extent SUEWS addresses the lateral water flow among different types.

Based on literature, in SUEWS, the surface water lateral flow within the grid cell, but among the different land use types, is simulated by the following steps:

Each surface calculates its drainage based on the water balance equations. The water distribution matrix, which specifies how water is redistributed among different surface types, is used for the surface lateral flow process. For example, water drained from a paved surface might be partially redirected to nearby grass or soil surfaces. When the infiltration capacity is exceeded, excess water becomes surface runoff. The model calculates potential runoff for each surface type and redistributes it according to the water distribution matrix.

Besides, SUEWS accounts for both above-ground (surface runoff) and below-ground (subsurface flow) water movements. This ensures a comprehensive simulation of lateral water flow. We included this part in section 5.4.1 and Table 7.

LL 255. Please note that a main advancement of UT&C is the capability to consider physiological and biophysical properties of vegetation, and thus being able to consider different vegetation types – at least in principle, which was not the case in any of the other models except partially VTUF-3D

Thank you for your valuable insight on the model UT&C. We added the mentioned main advancement of the UT&C in the manuscript:

*To further assess the impact of vegetation on urban climate and hydrology, Meili et al. (2020) presented an urban eco-hydrological model, Urban Tethys-Chloris (UT&C). The development of UT&C combines components of the eco-hydrological model Tethys-Chloris (T&C) and ASLUM, thus including more detailed hydroclimatic processes. Besides, UT&C has the capability to consider the physiological and biophysical properties of vegetation, and thus can consider different vegetation types, at least in principle, which was not the case in any of the other models, except partially in VTUF-3d (Meili et al., 2020, 2025).*

LL 261. They take "wind speed" too as input. I suggest having the complete list of six environmental variables. Pr, Ta, RH, Ws, Rsw, Ldw, some models need atmospheric pressure too.

Indeed, some models take wind speed as input forcing. We revised the sentences and the Figure 2.

LL 276. The second acronym should be TUF-3D, without "V".

We revised this. Thank you for checking.

LL 313 and below. Maybe using "grid cell" rather than "square" would be more aligned with the previous literature. Overall, I think this part can be shortened.

We modified the manuscript using the "grid cell". And also make this paragraph shorter, showing together with the response in the following comment.

LL320-333. I find all this part a bit convoluted and not very clear, please consider rewriting and probably shortening, as it might not be so essential.

The paragraph is shortened and more concise:

*Figures 3c and d show the structures of the coupled mode in climate models, which simulate the entire study area in one go. These models use square grid cells of uniform size. SUEWS, similar to a complete land surface model, includes impervious land. Recently, WRF-SUEWS (Sun et al., 2023) was developed as a land surface scheme. The main difference between standalone and coupled SUEWS is the interaction of atmospheric conditions with the land surface. The coupled mode uses Urban Canopy Models (UCMs) for impervious areas, with the fraction set manually based on urbanization or land use data. The land surface model handles other landscapes like grass or bare land. Interactions between impervious and other land areas occur indirectly through air exchange. Coupled mode simulations resolve grid-to-grid atmospheric differences but limit grid-to-grid interactions. Coupled mode 2D UCMs focus on impervious land, while standalone UCMs integrate various urban canyon processes (Ryu et al., 2016; Wang, 2014; Lemonsu et al., 2012). Thus, standalone models are usually more advanced and comprehensive for urban process (Krayenhoff et al., 2020; Meyer et al., 2020).*

LL346-347. Please note that "urban block scale" and "neighborhood area" are quite vague and actually they might largely overlap, a clarification about scale might be helpful here.

We acknowledge that the terms "urban block scale" and "neighborhood area" have a significant overlap. We changed the description enhancing the precision and clarity.

*Consequently, UCM models are more widely utilized for studying regional impacts at the hundred-meter scale, and high-resolution CFD models are primarily used for detailed research at the meter scale.*

LL 350. "present" rather than "utilise"

We revised this.

LL 402-403. Please note that data availability might be an issue when defining branching and drainage structures in many urban environments, as sewage system locations are often poorly known in old cities or not available in modern cities. This can also be discussed later.

Indeed, we totally agree that the availability of data is an essential issue. We included this in the challenge and future development section of the manuscript.

In Section 6 we have a discussion on this:

*However, both 3D ULSMs and URBS hydrological models require high-resolution input data, such as urban morphology, topography, and sewer systems. This need for detailed urban data limits the application of URBS compared to models like SWMM. As data collection and measurement methods for urban morphology and infrastructure improve, high-resolution simulations of urban microclimates and hydrological processes at the neighborhood scale may become feasible, enabling comprehensive thermal and hydro-hydraulic interactions.*

LL 435. Are you referring to water surface as "urban lakes" or "ponding water in various surfaces" the latter is likely solved in most U-LSMs.

We are referring to urban surface waters, including urban lakes. We revised the sentence to be clearer. Our review indicates that most Urban Land Surface Models (U-LSMs) do not incorporate these urban surface water. Specifically, only the SUEWS, TARGET, and Solene-Microclimat models resolve the energy balance above urban surface water. In most Urban Canopy Models (UCMs), only canyon elements (streets and buildings) and canyon vegetation are represented, meaning surface waters are not addressed within an urban grid. In coupled models, where ULSMs are integrated with regional climate models, all water surfaces are simulated solely by the Land Surface Model (LSM). Our study highlights the need to consider how urban surface waters, which can differ significantly from natural landscapes, influence the urban micro-mesoscale climate.

LL 458. Please note that the shortwave and longwave radiation budget is different and not only sky view factors but view factors among different urban facets are required, at least for longwave exchange, it is also not true that all U-LSMs do not compute multiple reflections.

We adjusted the sentence.

The phrase "radiosity method" is directly adopted from the literature on the Solene-Microclimat model. The "radiosity" is not an English term, but a French term. Thus, we put the quotes for it to make it clear.

We also have an explanation of how this method works: "*This method involves two critical aspects: 1) geometric form factors between all facets of the built surface and the sky vault are calculated using the contour integral technique; 2) radiative properties (reflection, transmission, and absorption) are considered for all surfaces within the scene.*" We revised this part of the manuscript and make the description of this method clearer.

Table 5. Maybe it could be interesting subdividing between models using a multi-layer vs two- or three-layers approach for solving the urban canyon, as more than one layer is not lumped anymore.

We agree that more than one layer is not lumped. In Table 5, if the model can solve more than one layer value within the canyon, we marked it as P (profile). We double-checked all the models to what extent they solve the hydro-meteorological conditions and distinguish the lumped (single layer), two layers and multi-layer (three and above) models. There are only single and multi-layer models in our selected models. Thus, we keep the Lump (L) and Profile (P) for the table.

LL 495. "area" rather than "ares".

We corrected this.

LL 537. Urban Hydrological Models not "Urban Heat Models", which does not mean anything.

We revised this.

LL 598. Applying the 1d Richards equation allows for the solution of infiltration, but it is much more than an infiltration method, as it allows the solution of variably saturated flow in multiple soil layers.

Indeed, we agree with the comment. In Section 5.4.1, we mentioned that the vertical moisture transport in multiple soil layers is solved by the 1D Richards equation in SLUCM, TEB, UT&C and VUCM.

LL 604-605. The description of the "Horton method" is wrong. It is the other way around, infiltration decreases exponentially up to an asymptotic constant value at large times, which is typically the saturated hydraulic conductivity.

We double-checked the Horton method, and we agree with your comments. We revised the manuscript:

*The Horton method is an empirical model. It assumes that infiltration decreases exponentially over time until it reaches a constant value, typically the saturated hydraulic conductivity.*

LL 615. Please note that on most impervious surfaces (e.g., roofs, paved streets), the saturated hydraulic conductivity will be fundamentally zero, so why one should compute infiltration? This is not clear to me.

You are correct that infiltration over impervious surfaces is negligible, which is consistent with the approach taken by most ULSMs. Our intention is not to suggest that models should compute infiltration over impervious land. Rather, we are highlighting that some models set a constant rate for infiltration over impervious surfaces, while others do not. This is a summary statement. We made it clearer in the manuscript and propose the following sentences:

*Generally, if a model includes the water balance, it considers depression storage and infiltration. Based on the above summary, depression storage over impervious land areas can be solved relatively easily. The infiltration process over impervious land is not taken into consideration because the saturated hydraulic conductivity of these surfaces (e.g., roofs, paved streets) is fundamentally zero.*

LL623. Do you mean Table 7?

Yes, we revised it.

LL 638. In VTUF-3D and UT&C transpiration is indeed simulated using an explicit solution of stomatal conductance, which is a function of photosynthesis, so plant photosynthesis needs to be solved.

We expanded our description of the process of solving the transpiration in VTUF-3D and UT&C.

LL646-656. I think here, there is a bit of confusion around Penman-Monteith equation method. If a model solves the energy budget (i.e., latent heat) by solving for surface temperature/s, there is no need to use Penman-Monteith equation, which is actually a simplification of the overall energy budget and it does not guarantee energy budget closure. I think which models use Penman-Monteith and which aren't should be better described and specified.

We summarised this and made it clear in the manuscript.

There are several groups of methods to calculate the evaporation:

1) SUEWS (Ward et al., 2016) and WEP (Jia et al., 2001): calculated based on the Penman-Monteith equation. The models first obtains the net radiation, equal to the sum of the heat fluxes, including sensible, latent and ground heat fluxes. The storage heat flux and anthropogenic heat flux are calculated, and then the latent heat flux is calculated based on the Penman-Monteith equation. The energy balance closure is solved by setting the residual as the sensible heat flux.
2) VUCM, ASLUCM, TEB, UT&C, VCWG (Lee, 2011; Meili et al., 2020; Masson, 1999; Wang et al., 2013): solve the evaporation using the resistance method. This method directly calculates the moisture heat flux based on the saturation value of specific humidity at a given surface temperature. These models directly solve the energy balance and the surface temperature.
3) URB (Berthier et al., 2004): The evaporation flux is assumed to be proportional to the potential evapotranspiration and water storage, while the potential evapotranspiration is the model's input forcing.
4) SWMM (Rossman,2017): three simplified ways, including setting as a single constant value, getting from the input data, and computing the daily values from the daily temperatures in an external climate file

LL 692-694. Does this refer to canopy or ground vegetation? This is not clear.

We double-checked the literature Lee and Park, 2008. The vegetation referred to in the study is canopy vegetation. We made it clear in the manuscript.

LL723. The end of the sentence is not clear, which empirical models? Which calibration?

We revised this part of the manuscript with more details as follows:

*Soil conductivity and saturated soil conductivity in urban hydrological models are consistent with those represented in most Urban Land Surface Models (ULSMs). These parameters can be calibrated in the field using infiltrometers and laboratory measurements, such as the constant head and falling head methods (Gupta et al., 2021). Additionally, they can be estimated using empirical models based on soil type, moisture content, and temperature, such as Pedotransfer Functions (Van Looy et al., 2017). The accuracy of these estimates can be further enhanced through machine learning techniques, including artificial neural networks and random forests (Jian et al., 2021).*

LL770-772. I am not sure I understand the question here and especially why it is formulated as a question. It is mostly a parameterization issue how to deal with these processes.

We revised the sentence:

*As stand-alone ULSMs increasingly incorporate natural land surfaces and interaction processes, it becomes imperative for regional atmosphere-land models and ULSMs to*

*systematically address the radiation, aerodynamic, and dynamic interactions between natural landscapes and anthropogenically influenced landscapes, particularly water surfaces and built-up areas.*

LL 778. I am not sure I understand why ones would like to have two-ways feedback from the hydraulic part to the atmosphere, being the flood response mostly a very fast process and occurring largely in impermeable channels and drains.

We understand your perspective on the rapid nature of flood response and the role of impermeable channels and drains. Indeed, the atmospheric simulation output can be the input for the urban flooding forecast. However, we believe there are several reasons why linking atmospheric models, land surface models, and hydrological-hydraulic models can be beneficial: 1) Two-way feedback allows for more precise monitoring and control of hydraulic systems, which can improve the accuracy of flood predictions and responses. Currently, the hydraulic model takes a single grid precipitation data as the input. However, with a deeper understanding of the urban impact on precipitation events, the spatial variability can be quite large, even within a catchment, depending on its size. Thus, higher-resolution input data for hydraulic systems can provide better performance. 2) Two-way feedback can help manage waterlogging more effectively by providing real-time data on soil saturation and atmospheric conditions. This can lead to better predictions and responses to waterlogging events. 3) Two-way feedback can improve the simulation of the urban micro-climate during and after extreme precipitation events that may lead to urban flooding. This can also optimize drainage and reduce the negative impacts on local urban ecosystems.

To be clearer, more accurate and specific, as also suggested by reviewer 3, we revised our manuscript for Section 6, Challenges and future developments, to include a more detailed explanation of the necessary model coupling. The revised Section 6 explicitly frames future developments around specific urban hydroclimate challenges, including urban thermal environment adaptation, urban flood forecasting, and compound extreme event analysis. The necessity and significance of model coupling strategies and modelling approaches are addressed for each specific topic.

LL 875. As the authors know, total evaporation and latent heat flux is the same thing, just different units, the sentence is awkward.

Yes, we revised the sentence:

*The surface water balance in ULSMs integrates evaporation, which is expressed as latent heat flux, thereby linking the water and energy balances. Conversely, UHMs may neglect or aggregate evaporation processes. ULSMs typically include multiple subsurface soil layers with vertical water transfer, predominantly within the unsaturated zone. However, their description of the saturated zone is often limited.*

**References**

Meili, N., Zheng, X., Takane, Y., Nakajima, K., Yamaguchi, K., Chi, D., Zhu, Y., Wang, J., Qiu, Y., Paschalis, A. and Manoli, G., 2025. Modeling the effect of trees on energy demand for indoor cooling and dehumidification across cities and climates. Journal of Advances in Modeling Earth Systems, 17(3), p.e2024MS004590.

Comments from Reviewer 3:

This manuscript is a timely and valuable review of urban land surface and hydrology models, bringing together diverse modelling approaches. The authors successfully highlight both achievements and persistent gaps in the field. The paper's organisation is logical and the narrative engaging, providing a thoughtful synthesis across historically fragmented modelling communities - I thoroughly enjoyed reading this review paper.

I believe, however, the paper would benefit from addressing two key aspects that would further enhance its clarity, structure, and practical utility.

Dear Reviewer:

Thank you for your thoughtful and encouraging comments on our manuscript. We greatly appreciate your recognition of our efforts to highlight achievements and persistent gaps in the field.

We also appreciate your constructive feedback regarding the two key aspects that could enhance our work. We are committed to improving our manuscript and have carefully addressed these points in our revisions with a new schematic diagram and revised discussions of future directions.

Best regards,

Xuan and co-authors

Major Comment 1: The paper would benefit from a conceptual schematic and stronger directory-style guidance, ideally integrated into Section 2.

While the manuscript does mention its rationale for selecting representative and newly developed models and those at the climatology-hydrology interface, it currently lacks an overarching conceptual framework that visually summarises the overall structure of the review. Including a schematic diagram early in Section 2, clearly illustrating how different urban processes (such as radiation, evapotranspiration, runoff, and soil moisture) relate to the various model classes reviewed (bulk, SL-UCM, ML-UCM, hydrology-focused models) and coupling strategies, would provide valuable directory-like guidance. Such a visual roadmap would help readers, especially those less familiar with the fxield, to better navigate and contextualise the rich and diverse content of subsequent sections.

We incorporated this conceptual framework of the overall structure to provide readers with a clearer understanding of the connections and context of the models reviewed. We incorporated the framework at the end of Section 1 while explaining the following structure and contents.

[Figure]

Major Comment 2: The future directions section should be more closely tied to specific scientific tasks and supported by clear technical mechanisms.

Currently, the call for integration and collaboration in Section 6 is valuable but somewhat general. To strengthen its practical impact, the authors could explicitly frame future development around specific urban hydroclimate challenges (e.g. urban flood forecasting, heat mitigation, compound event analysis), clarifying the necessary coupling strategies and modelling approaches.

Two concrete, actionable suggestions could help operationalise this vision:

Develop a common modelling protocol—similar to the NetCDF CF conventions widely used in climate sciences—which is currently missing for urban hydroclimate modelling. Such a protocol could standardise inputs, outputs, metadata, and resolution criteria to greatly facilitate interoperability and cross-model comparisons. Initial steps toward this goal have already been taken within the Urban-PLUMBER initiative, but these efforts should be expanded and formalised into more broadly accepted community practices.

Establish a practical technical framework to overcome barriers in collaborative model development. Leveraging open-source collaboration platforms like GitHub could foster transparency, modular development, and community engagement. Additionally, considering WRF's wide adoption and flexibility, it might serve effectively as a common base framework to host and test integrated urban hydroclimate components.

These steps would provide concrete pathways to improve the current fragmented landscape, enabling more coherent and coordinated model advancements.

We concur that focusing future development on specific urban hydroclimate challenges could significantly enhance the practical impact of our call for integration and collaboration. We revised our manuscript for Section 6, Challenges and future developments, to include a more detailed explanation of the necessary model coupling. The first three parts of the revised Section 6 explicitly frame future developments around specific urban hydroclimatic challenges, including urban thermal environment adaptation, urban flooding forecasting, and compound extreme events analysis. The necessity and significance of model coupling strategies and modelling approaches are addressed for each specific topic:

[revised manuscript text omitted]

L213:  The authors state that "Wang et al. (2016) added an irrigation scheme to the model" (referring to SUEWS). This attribution is incorrect. The irrigation functionality was first introduced in Järvi et al. (2011), which documented the initial release of the Surface Urban Energy and Water Balance Scheme (SUEWS). The cited Wang et al. (2016) paper does not pertain to SUEWS development and focuses instead on vegetation cooling in desert cities. I suggest correcting this reference to reflect the accurate model development history.

Reply to comment on L213:

 We revised this part of the manuscript.